# A novel WD40-repeat protein involved in formation of epidermal bladder cells in the halophyte quinoa

Tomohiro Imamura[1✉], Yasuo Yasui[2], Hironori Koga[3], Hiroki Takagi[3], Akira Abe[4], Kanako Nishizawa[1], Nobuyuki Mizuno[2], Shinya Ohki[5], Hiroharu Mizukoshi[6] & Masashi Mori[1✉]

Halophytes are plants that grow in high-salt environments and form characteristic epidermal bladder cells (EBCs) that are important for saline tolerance. To date, however, little has been revealed about the formation of these structures. To determine the genetic basis for their formation, we applied ethylmethanesulfonate mutagenesis and obtained two mutants with reduced levels of EBCs (*rebc*) and abnormal chloroplasts. *In silico* subtraction experiments revealed that the *rebc* phenotype was caused by mutation of *REBC*, which encodes a WD40 protein that localizes to the nucleus and chloroplasts. Phylogenetic and transformant analyses revealed that the REBC protein differs from TTG1, a WD40 protein involved in trichome formation. Furthermore, *rebc* mutants displayed damage to their shoot apices under abiotic stress, suggesting that EBCs may protect the shoot apex from such stress. These findings will help clarify the mechanisms underlying EBC formation and function.

[1] Research Institute for Bioresources and Biotechnology, Ishikawa Prefectural University, 308-1, Nonoichi, Ishikawa 921-8836, Japan. [2] Graduate School of Agriculture, Kyoto University, Sakyo-Ku, Kyoto 606-8502, Japan. [3] Department of Bioproduction Science, Ishikawa Prefectural University, 308-1, Nonoichi, Ishikawa 921-8836, Japan. [4] Iwate Biotechnology Research Center, 22-174-4 Narita, Kitakami, Iwate 024-0003, Japan. [5] Center for Nano Materials and Technology (CNMT), Japan Advanced Institute of Science and Technology (JAIST), 1-1 Asahidai, Nomi-Shi, Ishikawa 923-1292, Japan. [6] Technology Development Group, Actree Co., 375 Misumimachi, Hakusan, Ishikawa 924-0053, Japan. ✉email: timamura@ishikawa-pu.ac.jp; mori@ishikawa-pu.ac.jp

As sessile organisms, plants produce specialized epidermal cells, such as trichomes and root hairs that enable them to adapt to various conditions. Trichomes can protect plants from both biotic and abiotic stresses[1], and root hairs aid in moisture and nutrient absorption by increasing root surface area[2]. Halophytes, which can grow in high-salt environments, have developed unique epidermal tissues to protect themselves from the adverse effects of high salinity[3]. The secretion of salt by salt glands, arguably one of the most remarkable features of halophytes, is characteristic of many species from various families[3–5]. Nonglandular cells called epidermal bladder cells (EBCs) also accumulate salt in these plants.

EBCs, which are large vacuolated cells with or without stalks, are present only in *Aizoaceae* and *Amaranthaceae*[5]. Bladder cells are huge (~1000× the volume of normal epidermal cells) and can accumulate high concentrations of salt in their vacuoles: for example, the vacuoles of *Mesembryanthemum crystallinum* bladder cells accumulate salt at concentrations of up to 1 M[6–8]. The molecular mechanisms involved in salt accumulation by EBCs have recently been reported in quinoa (*Chenopodium quinoa* Willd)[9,10]. A positive correlation between the amount of EBCs and salt stress tolerance was also reported in quinoa[11]; furthermore, quinoa plants in which EBCs have been artificially removed are less tolerant to salt[12]. These findings indicate EBCs are involved in relieving high salt stress in plants by accumulating high salt concentrations. There is also some evidence that EBCs accumulate plant pigments such as betalain and various

metabolites in addition to salts[6,12]. In *Chenopodium* spp., including quinoa, the population of EBCs is dense at shoot apices (Fig. 1 and Supplementary Fig. 1). These findings suggest that EBCs have other functions in addition to salt accumulation. Although physiological analysis has elucidated some aspects of EBC function[9,10], no studies to date have investigated how EBCs develop. Recently, some candidate genes related to EBC formation were selected in *M. crystallinum*[13,14]. However, because the key genes involved in EBC formation have not yet been isolated, the molecular mechanism underlying the formation of these important cells remains unknown.

Quinoa is a halophyte pseudocereal that originates from the Andean highlands and exhibits high levels of tolerance to various environmental stresses[15]. The high nutritive value of quinoa seeds[16] has led the Food and Agriculture Organization of the United Nations to classify it as a potentially attractive crop[17]. In quinoa, EBCs composed of a single stalk cell and single bladder cell[18] are present at high densities in shoot apex, young leaves, and ears. Quinoa EBCs are involved not only in salt tolerance but also in resistance to ultraviolet (UV-B) stress[12,19]. Recently, our group and others sequenced an allo-tetraploid quinoa genome[9,20,21] and conducted gene functional analysis using the acquired data[22,23]. Furthermore, we generated quinoa mutants using ethylmethanesulfonate (EMS) and identified the genes responsible for several of the induced mutations[22]. Thus, molecular genetic analysis of quinoa is now possible.

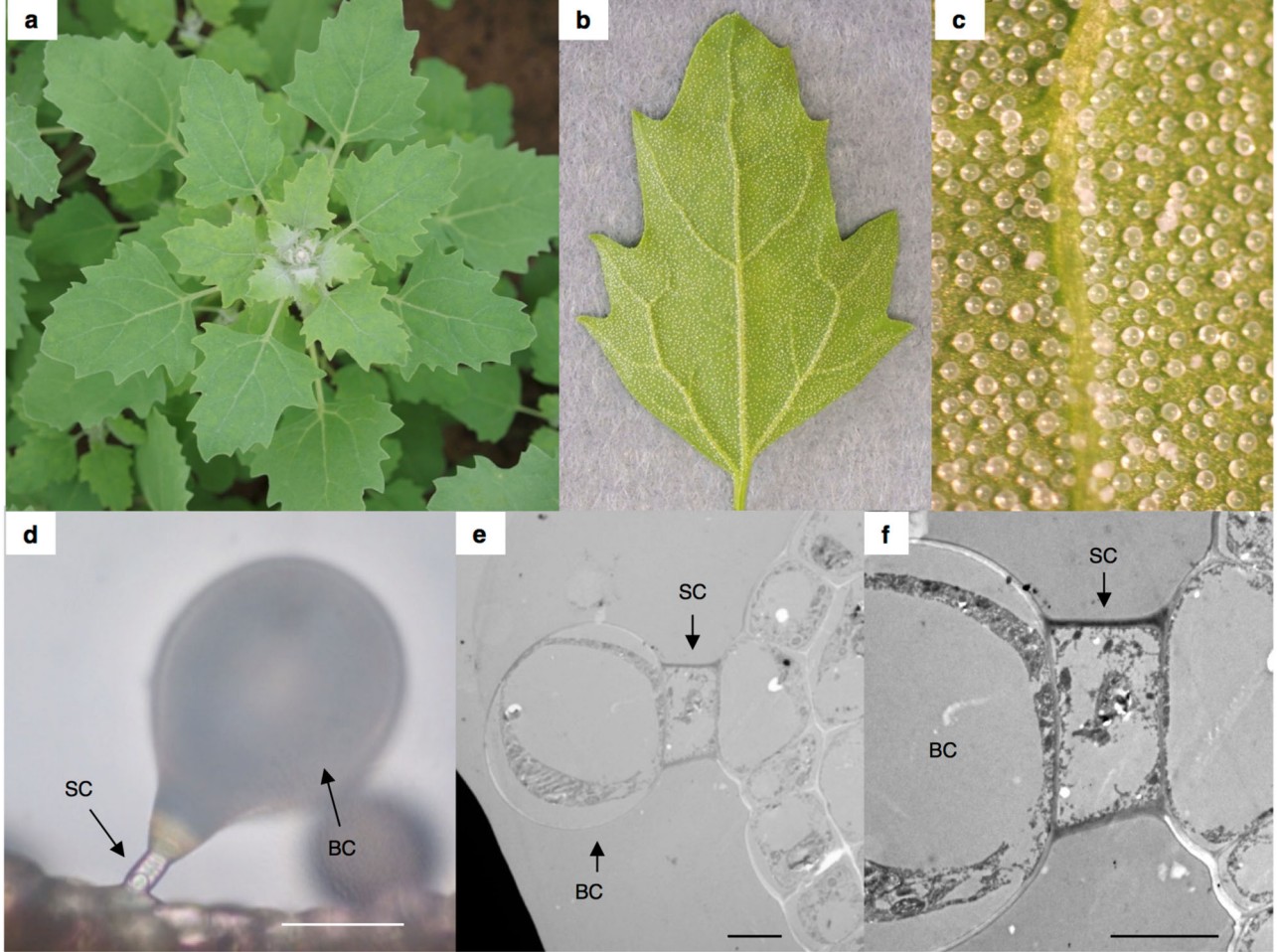

**Fig. 1 Epidermal bladder cells (EBCs) in quinoa.** One-month-old quinoa plants (**a**). Image of a young quinoa leaf (**b**). Close-up view of a young quinoa leaf (**c**). Images of a bladder cell (BC) and a stalk cell (SC) (**d**). Bar 50 μm. TEM image of an EBC (**e**). Bar 5 μm. **f** Close-up view of (**e**). Bar = 5 μm.

In this study, we sought to isolate genes involved in EBC formation and identify novel EBC functions. We performed EMS mutagenesis of quinoa to generate mutants defective in EBC formation, manifested as reduced number of EBCs (rebc). We succeeded in isolating genes involved in rebc mutation using an in silico subtraction method. Furthermore, we found that shoot apices of rebc mutants lacking EBCs were more severely damaged under abiotic stress.

## Results

**Observation of EBCs.** Individuals of several *Chenopodium* species were observed, and EBCs were identified in all cases (Fig. 1 and Supplementary Fig. 1). The populations of EBCs were so dense near the shoot apices of the plants that they obscured the shoot apex in all species studied (Fig. 1 and Supplementary Fig. 1). The EBCs of quinoa were examined in detail using both light microscopy and transmission electron microscopy (TEM). As previously reported[18], EBCs were composed of two cell types, namely, stalk and bladder cells (Fig. 1d–f).

**Production of mutants to study EBC formation.** EMS mutagenesis was conducted on approximately 8000 seeds of the CQ127 variety of quinoa. Two mutants (rebc1 and rebc2) with significantly reduced numbers of EBCs were isolated from the $M_3$ progeny (Fig. 2). In rebc mutants, the numbers of EBCs were <0.5% of those in the wild type (WT) (Supplementary Table 1). The outlines of young leaves and petioles were visible around the shoot apex in rebc mutants because the EBCs were absent (Fig. 2h, i). The mutants exhibited no differences in their cotyledons; however, their leaf color was a slightly lighter green than that of the WT (Fig. 2b, c).

**Evaluation of abiotic stress tolerance in rebc mutants.** We conducted abiotic stress treatments that directly affected shoot apices using WT plants and the rebc mutants with the lowest levels EBCs. First, we conducted wind treatment. All WT plants grew normally following 4 weeks of wind treatment, whereas rebc mutants exhibited damage to the shoot apex and defects in new leaf formation (Fig. 3a–f and Supplementary Table 2). Furthermore, WT plants grew normally during field cultivation, but rebc mutant plants suffered the same damage as those subjected to artificial wind treatment (Fig. 3g–i). These results suggest that EBCs play a role in protecting the shoot apex from certain stresses.

We also evaluated the tolerance of WT and rebc mutants to other abiotic stresses, such as UV-B radiation, high salinity, drought, and high temperature. The plants were irradiated by UV-B from above for 4 h, and then the damage around the shoot apex was evaluated. WT plants, the shoot apices of which were covered by EBCs, exhibited less damage to the shoot apex and leaves than rebc mutants (Fig. 4a–g and Supplementary Table 2). In parallel, WT plants and rebc mutants were cultured under high salinity conditions for 3 weeks (Fig. 4h), and damage to the shoot apex, growth of the shoot, and $Na^+$ content of the plants were subsequently evaluated. No damage due to high salinity was observed around the shoot apex in either WT plants or rebc mutants (Fig. 4i–n and Supplementary Table 2). Although the WT plants grew faster than rebc mutants (Fig. 4o–r), no significant difference was observed in $Na^+$ content between the WT and rebc plants (Fig. 4s, t). To investigate the accumulation of salt in EBCs in salt-treated quinoa, we compared the $Na^+$ content of leaves with or without EBCs. In the leaves from which EBCs were removed, the $Na^+$ content was reduced by approximately 10% relative to leaves with EBCs (Supplementary Fig. 2), suggesting that salt accumulated in EBCs under this treatment.

These results suggested that $Na^+$ accumulates to higher levels in the cell of rebc mutants than in those of WT plants. Furthermore, WT plants and rebc mutants were evaluated under conditions of high-temperature stress (42 °C, 10 days) and drought stress (water withheld for 10 days). No difference was observed between WT plants and rebc mutants under either high-temperature stress (Supplementary Fig. 3 and Supplementary Table 2) or drought stress (Supplementary Fig. 4 and Supplementary Table 3).

**Identification of the gene responsible for the rebc mutant.** To identify the gene or genes involved in the rebc mutant phenotypes, we first determined that the segregation ratio of both rebc mutants and the WT was 1:3 using the parent line ($M_2$) of rebc mutants (Supplementary Table 4). This Mendelian segregation suggests that the rebc mutants were homozygous, with a recessive phenotype caused by a single-gene mutation.

To identify the gene involved in the rebc1 mutant phenotype, DNA from 25 WT plants and 25 rebc mutants from heterologous rebc1 were sequenced using the Illumina HiSeq X system (Supplementary Table 5). Although the MutMap method[24,25] could not successfully identify the gene in question (Supplementary Table 6), we obtained 6555 short reads containing WT pool-specific 37-mers using an in silico subtraction method with custom Python scripts[26,27]. After assembling these reads using Velvet[28], we obtained 983 scaffolds, 14 of which were retained by filtering using BLASTN. Of these 14 scaffolds, only one (196 bp) had a nucleotide site at which the WT pool was heterozygous for the WT and mutant alleles and the rebc1 mutant pool was homozygous for the mutant allele. This scaffold is present in the predicted gene Cqu_c00398.1_g001.1 in the Quinoa Genome DataBase[20]. The reference genome registered in the Quinoa Genome DataBase has low accuracy because it is a draft genome. Therefore, when a BLAST search was conducted on Cqu_c00398.1_g001.1 in the NCBI database, in which a highly accurate quinoa genome is registered, only one gene (XM_021859495) was found. We named this gene, which is responsible for the rebc1 mutant phenotype (Supplementary Fig. 5), "*REDUCED EPIDERMAL BLADDER CELLS*" (REBC). WT REBC encodes a protein of unknown function containing six WD40 domains. The WD40 domain is one of the most common protein–protein interaction domains in eukaryotic proteins and coordinates the assembly of multiprotein complexes[29]. Point mutations transformed the codons for tryptophan 380 and 131 in REBC into stop codons in the rebc1 and rebc2 mutants, respectively (Fig. 5a).

To date, no method for generating specific transgenic plants in *Chenopodium* spp., including quinoa, has been established. Although we attempted *Agrobacterium*-mediated transformation using *Rhizobium radiobacter* (*A. tumefaciens*), particle bombardment, and viral vector methods to create transgenic quinoa plants, all methods failed. Hence, to confirm the identity of the gene responsible for the rebc phenotype, we crossed rebc1 and rebc2 plants to evaluate the traits of the $F_1$ progeny. If the mutation in REBC caused the rebc phenotype, all $F_1$ plants from the cross between rebc1 and rebc2 would exhibit the rebc phenotype; if not, all of the $F_1$ plants would exhibit the WT phenotype. We found that all eight $F_1$ plants from crosses between rebc1 and rebc2 exhibited the rebc phenotype (Fig. 5 and Supplementary Tables 1 and 4), indicating that REBC is the mutated gene responsible for the rebc phenotype. Furthermore, we conducted transient complementation experiments using *Rhizobium rhizogenes* in quinoa rebc mutants. Infection of mutant plants with a line carrying a plasmid expressing WT REBC resulted in the formation of a few EBCs on leaves, whereas no EBCs formed on leaves infected with a control vector

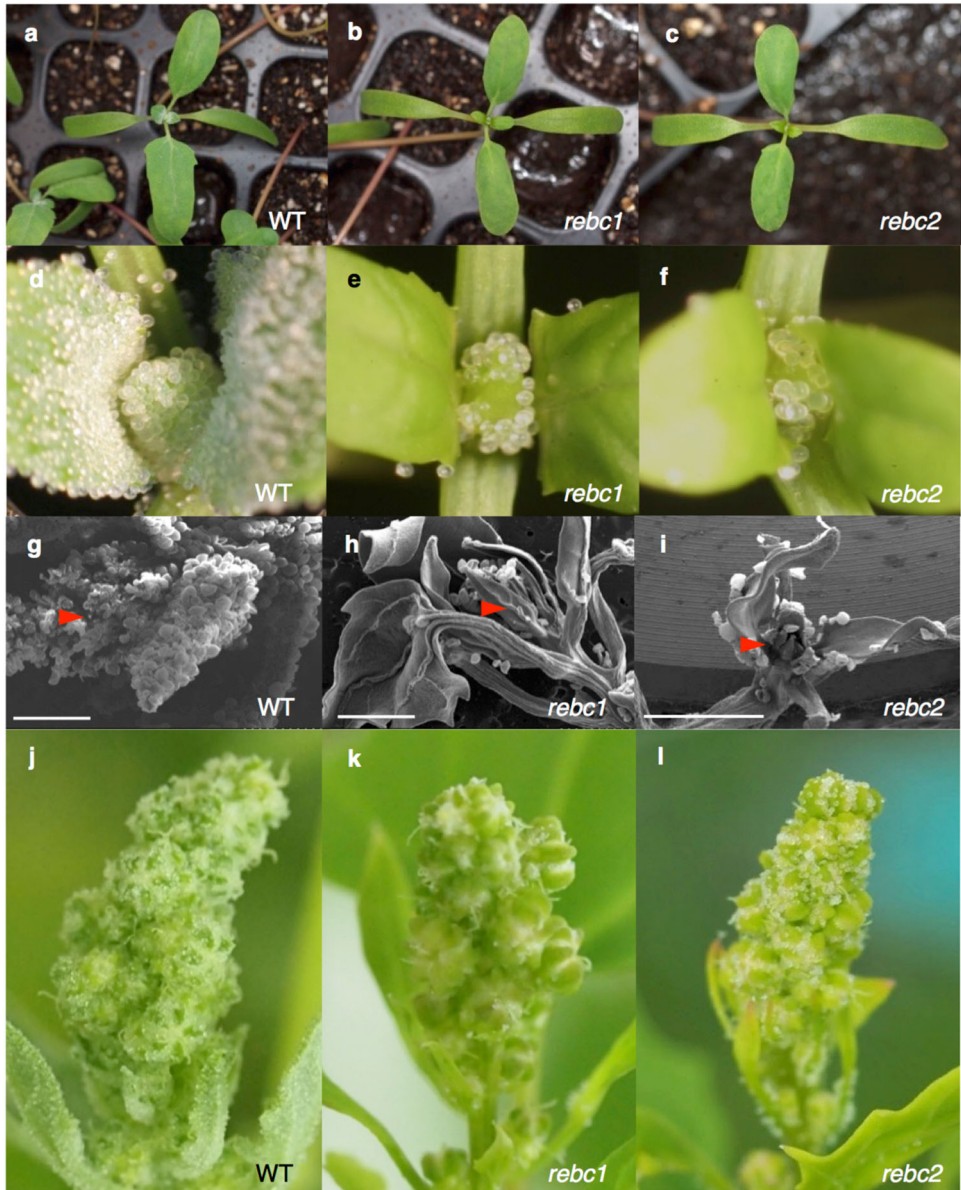

**Fig. 2 Phenotype of the *reduced epidermal bladder cells* (*rebc*) mutant.** Two-week-old quinoa seedlings of wild-type (WT; **a**), *rebc1* (**b**), and *rebc2* (**c**) plants. Shoot apices of WT (**d**), *rebc1* (**e**), and *rebc2* (**f**) plants. SEM image of the shoot apex of WT (**g**), *rebc1* (**h**), and *rebc2* (**i**) plants. Red arrowheads indicate the shoot apices. Bars 1 mm. Young ears of WT (**j**), *rebc1* (**k**), and *rebc2* (**l**) plants.

(Supplementary Fig. 6). Transient expression experiments were also conducted using *Rhizobium rhizogenes* in WT plants. No significant difference in EBC density was observed between WT *REBC*-expressing plants and vector control (Supplementary Fig. 7). Collectively, these results indicate that mutations in *REBC* are responsible for the *rebc* mutant phenotype.

**Expression analysis of the *REBC* gene family in quinoa.** *REBC* mRNA was expressed in leaves, hypocotyls, shoot apices, young ears, and roots of quinoa, but not in cotyledons (Supplementary Fig. 8a). Although *REBC* expression was confirmed in the roots, no significant difference was observed in roots or root hair traits between WT plants and *rebc* mutants (Supplementary Fig. 8b). *REBC* expression was light-dependent (Supplementary Fig. 8a).

**Analysis of the REBC protein.** Because quinoa is an allotetraploid, we considered the possibility that homologs of *REBC*

exist. BLAST analysis resulted in the identification of *REBC-like1* and *REBC1-like2* (Fig. 5a). Relative to REBC protein, 46 C-terminal amino acid residues were deleted in REBC-like1, and 224 N-terminal amino acid residues were deleted in REBC-like2 (Fig. 6a). The WD40 domain and C-terminal tail structure of the REBC protein suggest that its three-dimensional structure is that of a β-propeller (Fig. 6b). REBC-like1 protein, which is not associated with the *rebc* mutant phenotype, forms a β-propeller structure despite partial deletion of its C-terminal tail. The β-propeller structure functions as a scaffold for protein interactions, and the specificity of interacting proteins is determined by sequences outside the WD40 domain itself[30]. This suggests that the structure of the C-terminal tail is critical for REBC activity.

To determine the subcellular localization of REBC protein, we generated an antibody against the REBC C-terminal tail and confirmed its specificity for REBC protein produced in *Escherichia coli* and plant cells (Fig. 6a and Supplementary Fig. 9). Immunoelectron microscopy revealed that REBC was localized to

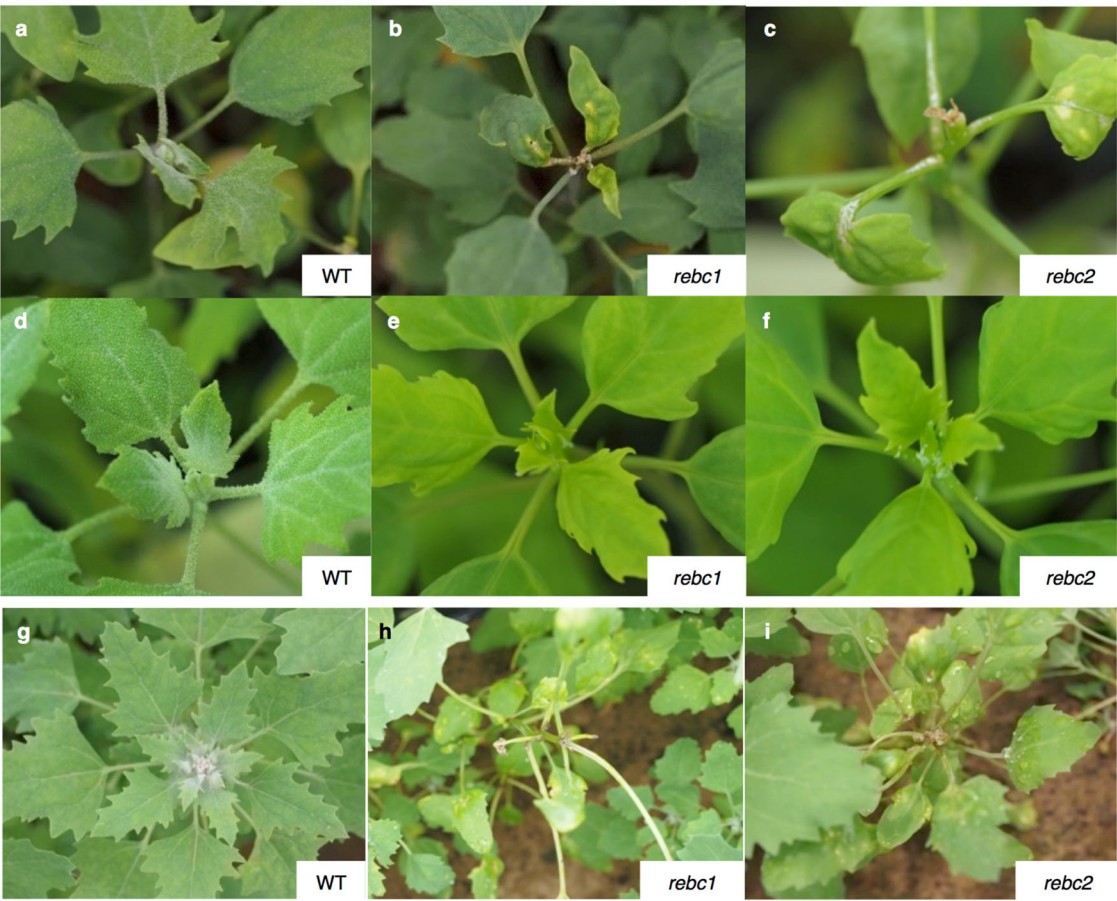

**Fig. 3 Wind treatment of quinoa.** Effects of 4 week of exposure to wind stress on WT (**a**), *rebc1* (**b**), and *rebc2* (**c**) plants. Non-treated WT (**d**), *rebc1* (**e**), and *rebc2* (**f**) plants. One-month-old field-grown quinoa WT (**g**), *rebc1* (**h**), and *rebc2* (**i**) plants.

the nuclei and chloroplasts in the leaves and to the nuclei in shoot apices (Fig. 6c–l). Because the *rebc1* mutant (negative control) lacks the C-terminal antibody-binding epitope, localization of the full-length the REBC protein in these mutants could not be determined (Fig. 6c–l). The WoLF PSORT algorithm (https://wolfpsort.hgc.jp/) also predicted that REBC would be localized to chloroplasts and nuclei.

**Comparative analysis of the REBC and TTG1 proteins**. The WD40 protein encoded by the *TRANSPARENT TESTA GLABRA1* (*TTG1*) gene in *Arabidopsis* reportedly plays a key role in trichome formation by epidermal cells[31]; however, the sequence similarity between the amino acid sequences of REBC and TTG1 was low (24% identity; Supplementary Fig. 10). No overexpression lines of *REBC* in the *Arabidopsis ttg1* mutant complemented the *ttg1* mutation (Fig. 7a, b and Supplementary Fig. 11). By contrast, quinoa has two orthologs of *TTG1*, *CqTTG1-like1*, and *CqTTG1-like2*, and all overexpression lines of *CqTTG1-like1* and *CqTTG1-like2* complemented the *Arabidopsis ttg1* mutation (Fig. 7a, b and Supplementary Fig. 11). Furthermore, phylogenetic analysis demonstrated that REBC and TTG1 belong to different functional groups (Fig. 7c). These results suggest that REBC belongs to a different group of WD40 proteins than TTG1 and is involved in the formation of EBCs.

**Comprehensive expression analysis in *rebc* mutants**. Next, we conducted RNA sequencing (RNA-seq) analysis to evaluate the gene expression profile of the *rebc* mutants (Supplementary

Table 5). We identified genes that were upregulated at least twofold or downregulated to ≤50% in *rebc* mutants relative to WT. Overall, 124 genes were downregulated in the two *rebc* mutants (Fig. 8a and Supplementary Data 1); this included several genes involved in disease responses, such as those encoding PR1 protein and antimicrobial protein (Supplementary Data 1). Furthermore, 115 genes were upregulated in the two *rebc* mutants (Fig. 8a and Supplementary Data 2). Recently, an RNA-seq analysis was conducted to evaluate gene expression in quinoa EBCs[9,10]. Using the published data, we investigated whether genes that were differentially expressed in our mutants were expressed in EBCs. In this analysis, gene expression was calculated using transcripts per kilobase million (TPM)[32] to facilitate comparison of our data with those of other groups. Out of 274 genes downregulated by at least 2-fold in the mutant, 123 were expressed in EBCs (Supplementary Data 3). Furthermore, 57 of 92 genes upregulated by at least 2-fold in the mutant were also expressed in EBCs (Supplementary Data 4). Next, we compared the genes with altered expression in *rebc* mutants were compared with those whose expression was altered in a mutant form of *M. crystallinum* that lacked EBCs[13,14]. A gene belonging to the jasmonate-induced protein family was downregulated in both *M. crystallinum* (WM28; NCBI Acc. No. KT366265) and quinoa (Phytozome Acc. No. AUR62022156) (Supplementary Data 3). Heterologous expression of *WM28* in *Arabidopsis* increases the number of trichomes[13]. Therefore, the quinoa jasmonate-induced protein gene and some of the differentially regulated genes in *rebc* mutants might be involved in EBC formation under the control of *REBC* gene in quinoa.

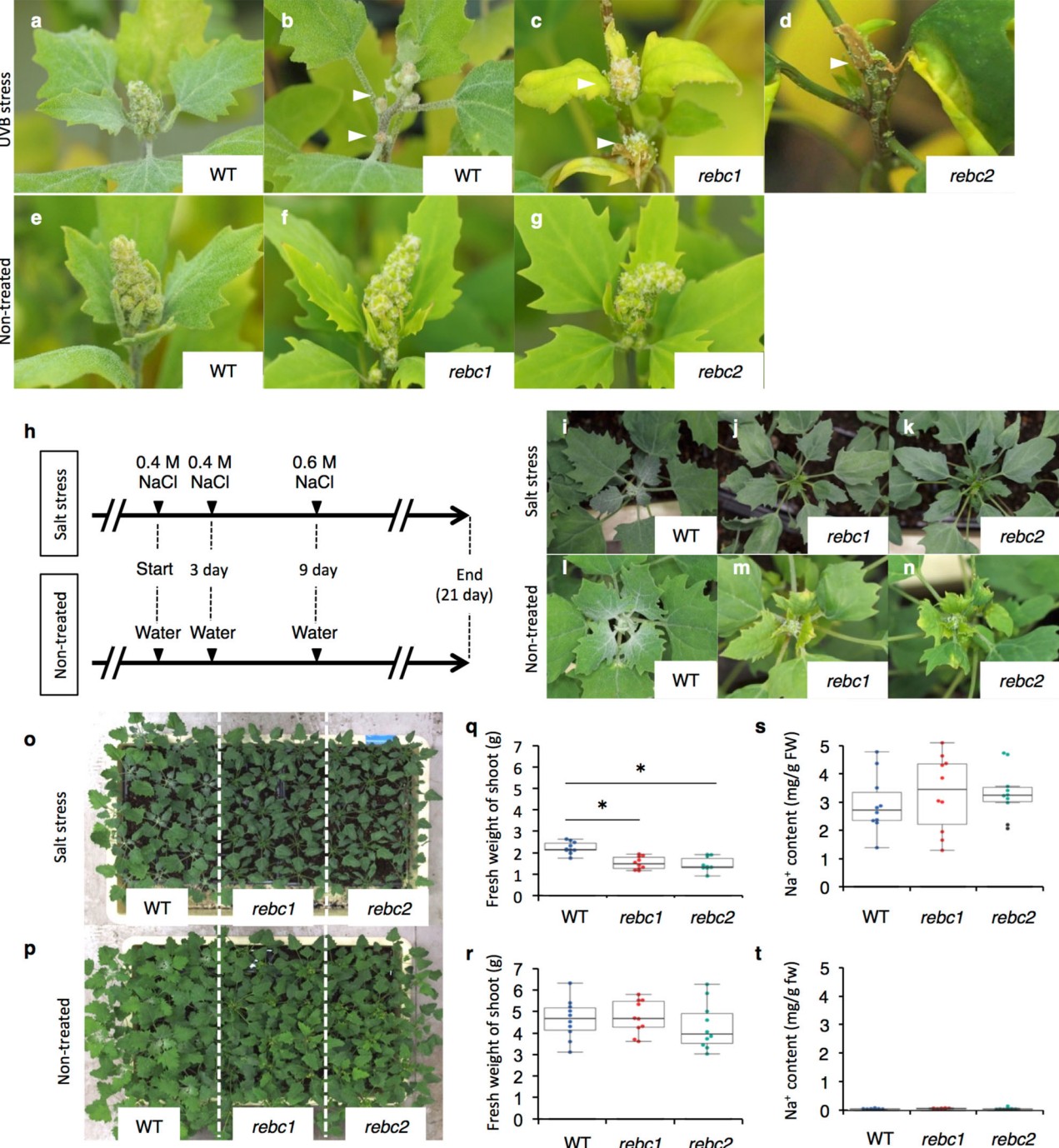

**Fig. 4 UV-B and high salinity treatment of quinoa.** Wild-type (WT; **a**, **b**), *rebc1* (**c**), and *rebc2* (**d**) plants after 2 weeks of UV-B treatment. Non-treated WT (**e**), *rebc1* (**f**), and *rebc2* (**g**) plants. White arrowheads indicate damaged shoot apices. Schematic diagrams of salt treatments in quinoa plants (**h**). Effects of 3 weeks of salt stress on shoot apices of WT (**i**), *rebc1* (**j**), and *rebc2* (**k**) plants. Non-treated shoot apices of WT (**l**), *rebc1* (**m**), and *rebc2* (**n**) plants. Photographs of salt-treated (**o**) and non-treated (**p**) plants. Fresh weights of shoots of salt-treated (**q**) and non-treated plants (**r**). Na⁺ contents of leaf lamina of salt-treated (**s**) and non-treated plants (**t**). Error bars represent the means ± SD; *$p < 0.05$ compared with WT. Data points are available in Supplementary Data 5.

**Observation of chloroplasts in *rebc* mutants.** Notably, numerous genes encoding chloroplast-localized proteins were either upregulated or downregulated in *rebc* mutants (Fig. 8a); we speculated that this would lead to observable alterations in the chloroplasts of these mutants. Hence, we measured chlorophyll contents and maximum quantum yield of fluorescence (Fv/Fm) to evaluate the function of chloroplasts in WT plants and *rebc* mutants. The chlorophyll contents were significantly lower in the *rebc1* and *rebc2* mutants than in the WT (Fig. 8b). By contrast, there was no significant difference in Fv/Fm, which is related to the activity of photosystem II (PSII), between WT plants and *rebc* mutants (Fig. 8c). Next, we measured the electron transport rate (ETR), which is also associated with PSII. The ETR of both *rebc* mutants was lower than that of the WT under strong light radiation, but not under weak light radiation (Fig. 8d). To clarify chloroplast morphology, we observed WT and *rebc* mutant

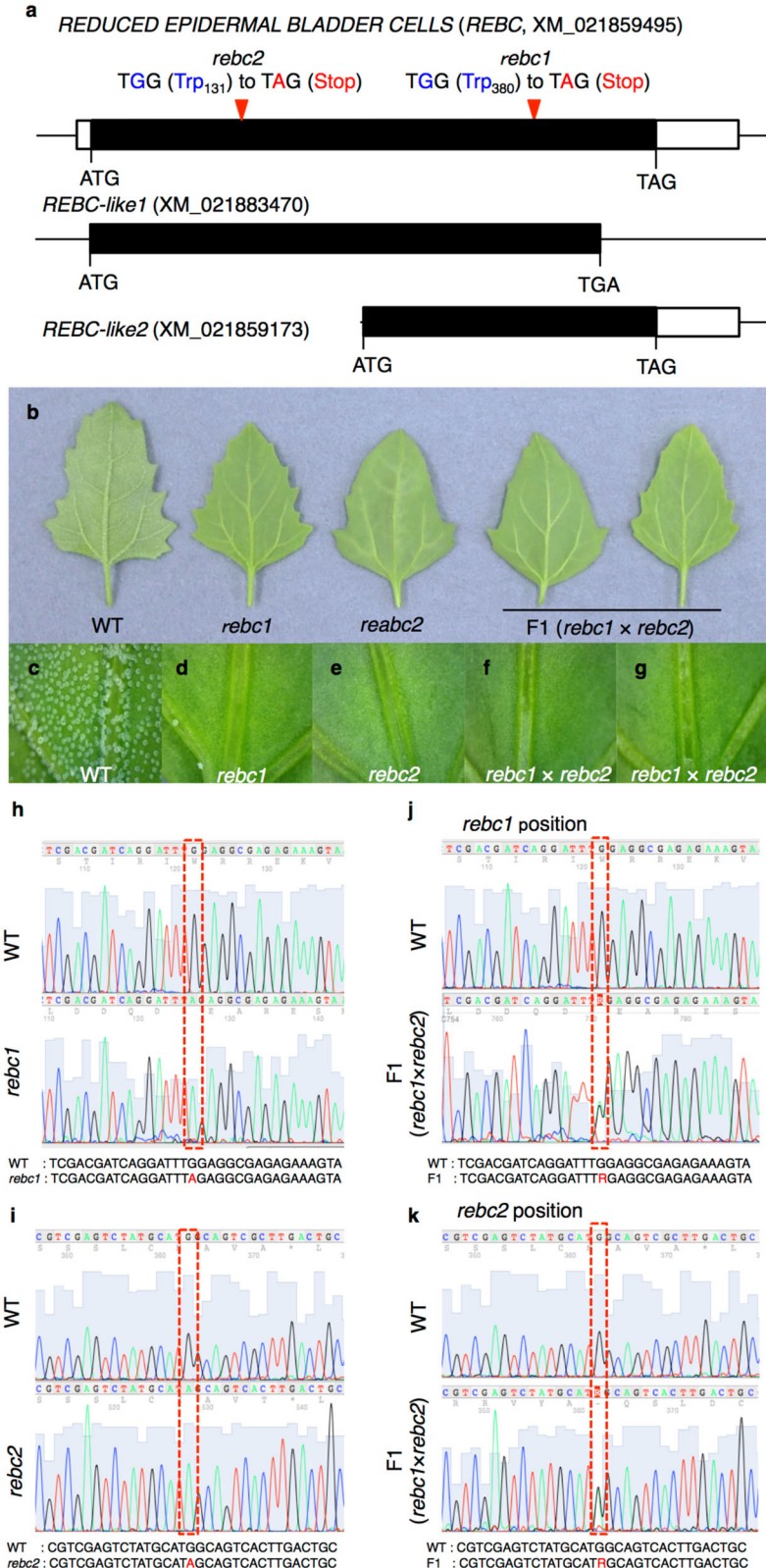

**Fig. 5 Identification of the gene responsible for the *rebc* mutant phenotype. a** Genomic structure of the *REDUCED EPIDERMAL BLADDER CELLS* (*REBC*) family. Open and closed boxes indicate untranslated and translated regions, respectively. Red arrowheads indicate positions of mutations. **b** Image of 1-month-old quinoa leaves. **c–g** Close-up views of (**b**). WT and *rebc* indicate wild-type and *rebc* mutants, respectively. **h** Direct sequencing of the *rebc1* mutant. Upper and lower panels show the sequence of WT and *rebc1*, respectively. **i** Direct sequencing of the *rebc2* mutant. Upper and lower panels show WT and *rebc2* sequences, respectively. **j, k** Direct sequencing of F₁ plants (*rebc1* × *rebc2*). Position of mutations in *rebc1* (**j**) and *rebc2* (**k**). Dashed red frames demarcate the mutation position.

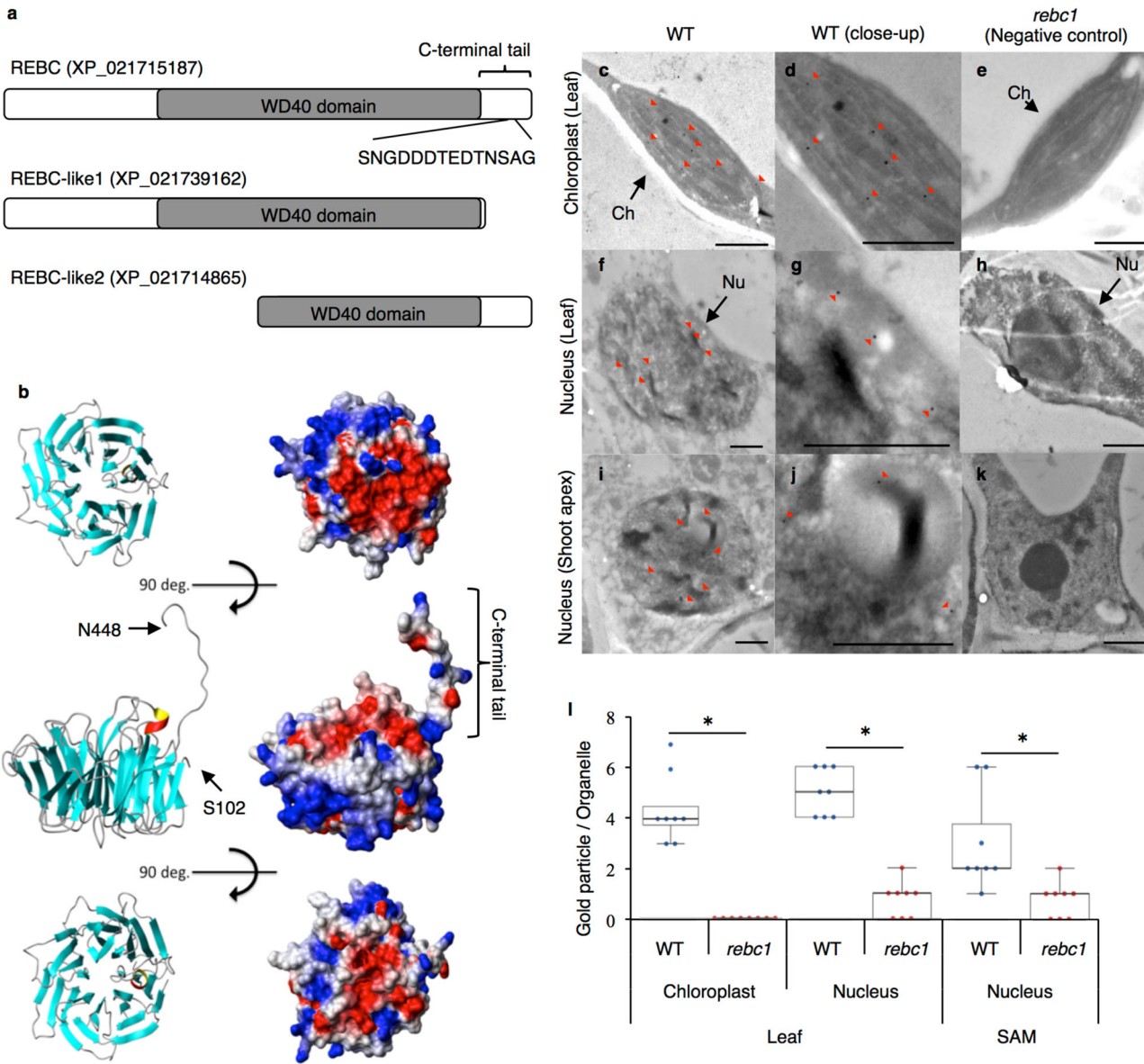

**Fig. 6 Molecular properties of the REBC protein. a** Schematic of the domain organization of the REBC family. The amino acid sequence is that of the antigen used for anti-peptide antibody production. **b** Three-dimensional structure of the partial REBC protein from S102 to N448. Left panels indicate secondary structural elements. Right panels show surface morphologies as determined by electrostatic potentials. **c–l** Subcellular localization of the REBC protein. Immunoelectron microscopy image of wild-type (WT; **c**, **d**, **f**, **g**, **i**, and **j**) and *rebc1* plants (negative control; **e**, **h**, and **k**). **d**, **g**, and **j** are close-up views of (**c**, **f**, and **i**), respectively. Red arrowheads indicate immunolabeled REBC protein. Nu, nucleus; Ch, chloroplast. Bars 1 μm. **l** Summary of REBC protein localizations. Error bars represent means ± SD; *$p < 0.05$ vs. WT. Data points are available in Supplementary Data 5.

chloroplasts using an electron microscope. Scanning electron microscopy (SEM) and TEM revealed that one-third of the cell wall-side lamellae were absent in all chloroplasts of *rebc* mutants cultured in the light (Fig. 8e–j and Supplementary Fig. 12). In quinoa, chloroplasts in bladder and stalk cells have reduced grana stacks[10] (Supplementary Fig. 13a–c). To investigate the detailed phenotype of the *rebc* mutant, we attempted to observe chloroplasts in *rebc* mutant EBCs. However, in *rebc* mutants, the number of EBCs was <0.5% of that in the WT (Supplementary Table 1); consequently, it was very difficult to observe the EBC chloroplasts of *rebc* mutants by TEM, and we were only able to obtain TEM image data for the *rebc2* mutant. No significant differences were observed in the structures of EBC chloroplasts between WT plants and *rebc2* mutants (Supplementary Fig. 13b, d, e), although the EBC chloroplasts of the *rebc2* mutant were smaller than those of the WT (Supplementary Fig. 13b, d, e).

Next, to observe EBC chloroplast traits, we examined chlorophyll autofluorescence in the WT and *rebc* mutants. The amount of chlorophyll autofluorescence in *rebc* mutants was significantly reduced relative to the WT (Fig. 8k–m, q). Furthermore, the intensity of chlorophyll autofluorescence was significantly lower in *rebc* mutants than in the WT (Fig. 8k–m, r). These results indicate that the *rebc* mutation influenced chloroplast formation in both leaves and EBCs.

## Discussion

In this study, we identified the gene involved in EBC formation and a novel EBC function in the halophyte quinoa. Mutagen-treated quinoa seeds gave rise to two mutants that exhibited chloroplast abnormalities in leaves and EBCs and contained significantly fewer EBCs than the WT. Under abiotic stress

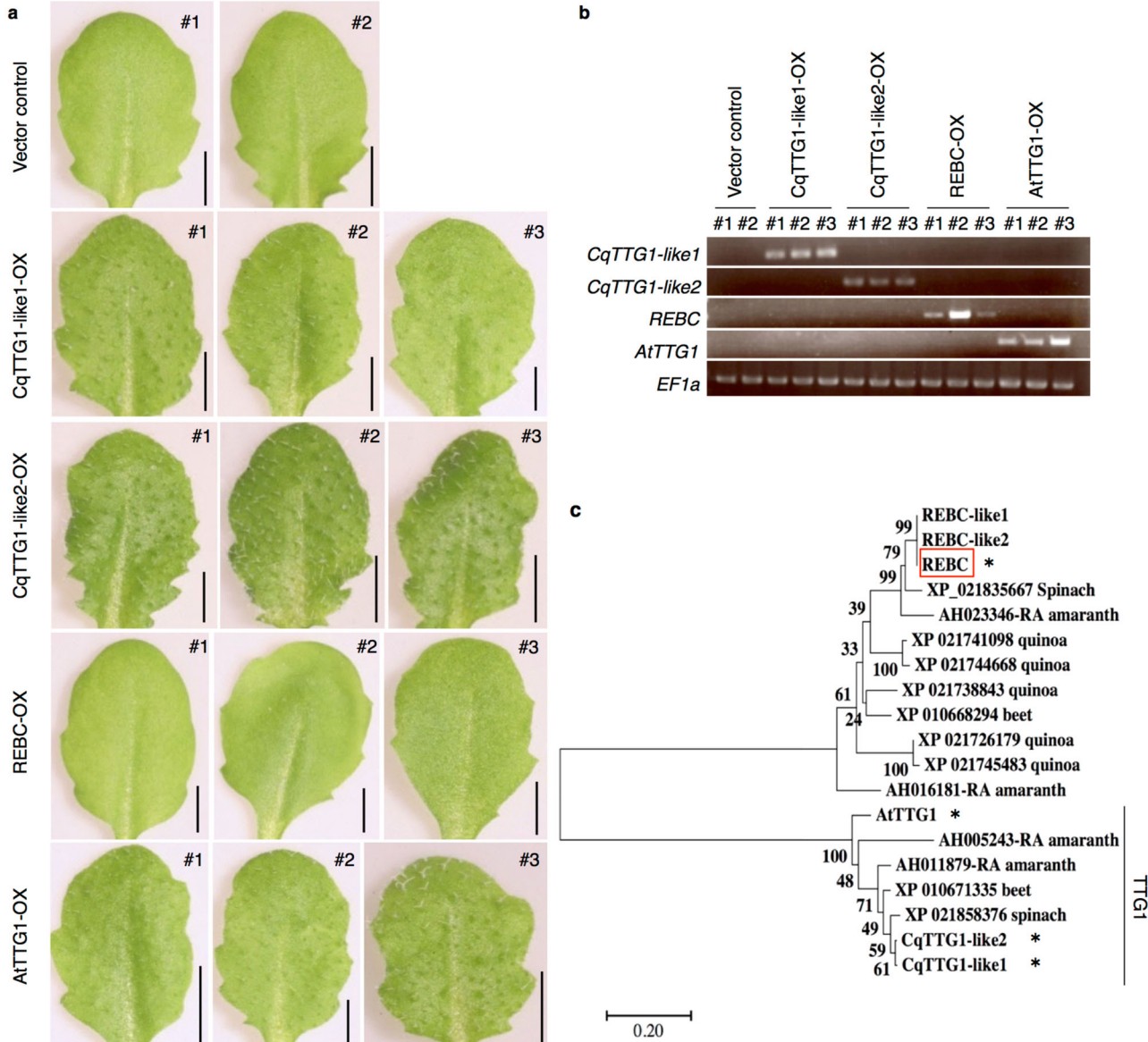

**Fig. 7 Comparative analysis of REBC and TTG1 proteins. a** Complementation test using the *Arabidopsis ttg1-21* mutant. The rosette leaves of a transgenic plant for vector control, the *CqTTG1-like1* overexpression line (CqTTG1-like1-OX), the *CqTTG1-like2* overexpression line (CqTTG1-like2-OX), the *REBC* overexpression line (REBC-OX), and the *AtTTG1* overexpression line (AtTTG1-OX) are shown. Bars 2 mm. **b** RT-PCR analysis of gene expression in transgenic plants. *EF1α* indicates an internal control. #1–#3 are the individual index numbers of these transgenic lines. **c** Phylogenetic analysis of groups highly homologous with the REBC and TTG1 families in *Amaranthaceae* (quinoa, spinach, amaranth, and beet) and *Arabidopsis*. The red frame indicates the REBC protein. Asterisks indicate proteins used for complementation testing. Branch lengths correspond to the divergence of sequences, as indicated by the scale on the lower left. Quinoa and spinach both form EBCs.

conditions, these *rebc* mutants displayed damage to their shoot apices. In silico subtraction experiments revealed that mutation of the *REBC* gene was responsible for the *rebc* phenotype. We showed that *REBC* encodes a WD40 protein that localizes to both nuclei and chloroplasts. Furthermore, phylogenetic and transgenic plant analyses revealed that the REBC protein differs from TTG1, which is involved in trichome formation, providing insight into the mechanism underlying EBC formation.

EBCs decrease saline-induced damage by accumulating salt[12] and protecting young leaves from UV-B stress[19]. The results of this study confirmed that EBCs are required for growth under high salinity and may protect the shoot apex from abiotic stressors, such as wind and UV-B, thereby helping plants grow normally in the field. The shoot apex is important for plant growth

and contains undifferentiated cells in its apical meristem, that require protection from environmental stress. In quinoa, EBCs in this area are dense and cover the shoot apex and this dense packing may protect the shoot apex from environmental stress. EBCs enlarge by endoreduplication[33], which allows the shoot apex to be protected even when only a small number of EBCs are present. A high density of EBCs at the shoot apex was also observed in *Chenopodium* spp, in which a high density of EBCs may protect the shoot apex and allow growth in harsh environments. Although not as dense as in quinoa, trichomes are similarly dispersed around shoot apices in *Arabidopsis* (Supplementary Fig. 14). When we conducted the same stress experiments in *Arabidopsis* as in quinoa, we observed no difference between the WT and a *ttg1* mutant, suggesting that

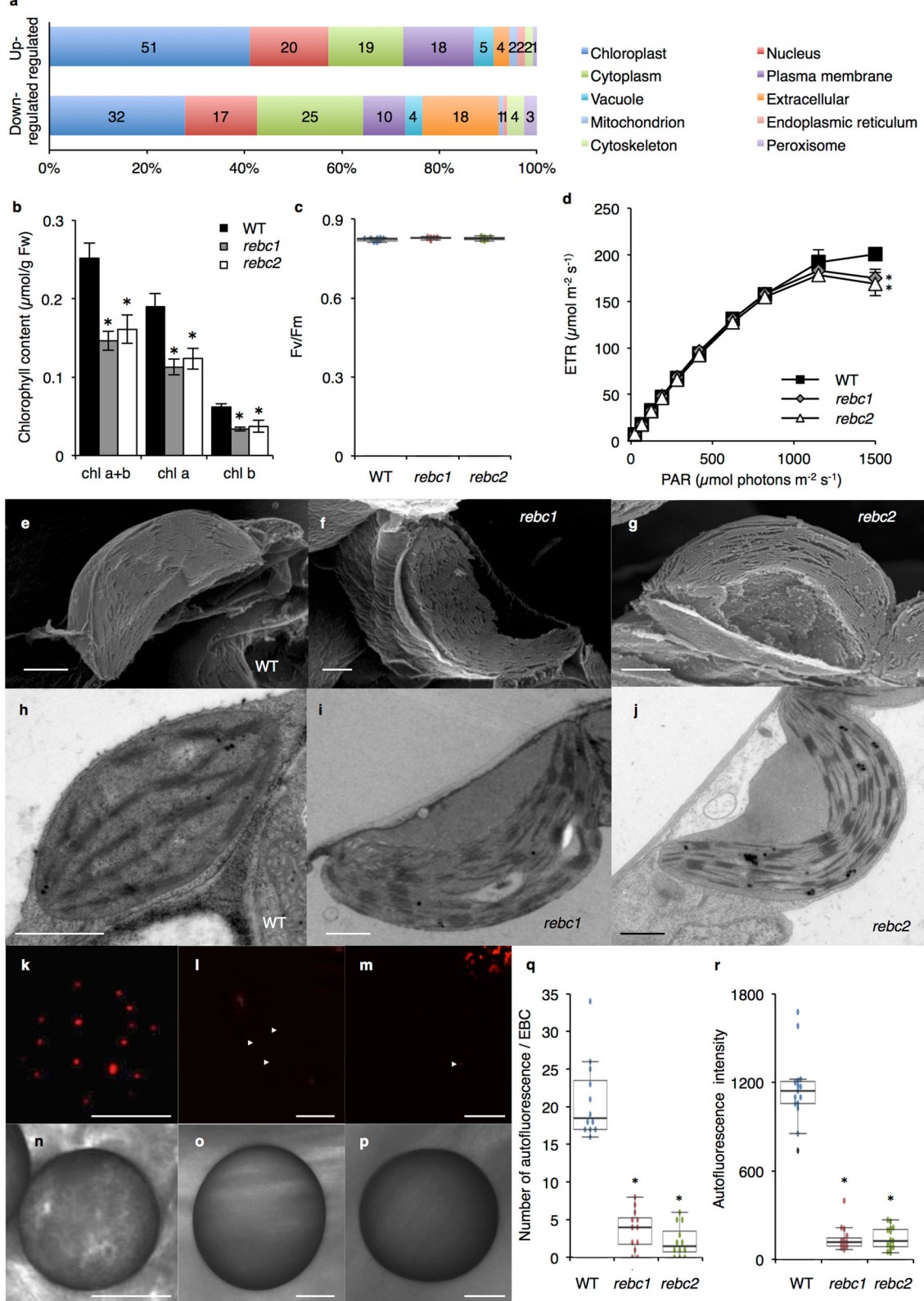

trichomes do not play a role in protecting shoot apices from environmental stress (Supplementary Fig. 14). Thus, protection of the shoot apex from environmental stress may be one of the unique functions of EBCs. In quinoa, it is better to cover the shoot apex completely with EBCs than to cover it with a trichome, which would leave a gap. Ensuring that the shoot apex is protected, even at the expense of the extra energy required for

EBC formation, enables quinoa to grow in harsh environments. Therefore, our results also reveal that EBCs function in protecting specific tissues, but not the entire plant, from environmental stress.

*REBC*, which is involved in EBC formation, encodes a WD40 protein. TTG1, also a WD40 protein, plays a central role in the formation of trichomes[31,34], which are epidermal structures

**Fig. 8 Chloroplast phenotype of *rebc* mutants. a** Genes with altered expression in the two *rebc* mutants. "Upregulated" and "downregulated" indicate that genes were expressed at ≥ twofold higher or lower levels in the mutant vs. the wild-type (WT). Boxes indicate the subcellular organelles to which the proteins are predicted to be localized. Numbers in the boxes indicate the number of gene products predicted to be localized to that organelle. **b** Chlorophyll contents of WT and *rebc* mutants. Black, gray, and white bars indicate the chlorophyll contents of WT, *rebc1*, and *rebc2* plants, respectively. **c** Fv/Fm ratio of quinoa plants. **d** Light response curves of ETR in WT and *rebc* mutants. Measurements were conducted at the following light intensities (photosynthetically active radiation, PAR): 25, 65, 125, 190, 285, 420, 625, 820, 1150, and 1500 μmol photons m$^{-2}$ s$^{-1}$. Data for WT (filled squares), *rebc1* (gray diamonds), and *rebc2* (open triangles) plants are shown. SEM images of leaf chloroplasts of WT (**e**), *rebc1* (**f**), and *rebc2* (**g**) plants. TEM images of leaf chloroplasts of WT (**h**), *rebc1* (**i**), and *rebc2* (**j**) plants. Bars 1 μm. Chlorophyll autofluorescence images of EBCs from WT (**k**), *rebc1* (**l**), and *rebc2* (**m**) plants. Arrowheads indicate chlorophyll autofluorescence in the *rebc* mutant. Bright-field images of EBCs from WT (**n**), *rebc1* (**o**), and *rebc2* (**p**) plants. Bars 50 μm. Amount of chlorophyll autofluorescence (**q**) and intensity of chlorophyll autofluorescence (**r**) in EBCs. Error bars represent means ± SD. *$p < 0.05$ compared with WT. Data points are available in Supplementary Data 5.

similar to EBCs. Although both proteins contain WD40 repeats, they are considered to be completely different proteins because they otherwise have low sequence similarity and *REBC* cannot complement the *ttg1* phenotype. Furthermore, *Arabidopsis* does not contain an ortholog of *REBC*. These findings suggest that EBCs form by different mechanisms than trichome. With the availability of next-generation sequencing, genetic information on EBC formation can easily be obtained. In this study, on the basis of RNA-seq analysis, we identified genes that could be involved in EBC formation. The identification of *REBC* will help elucidate the mechanisms of EBC formation in halophytes in future studies.

In quinoa, two *TTG1* orthologs (*CqTTG1-like1* and *CqTTG1-like2*) are present, and both can function as *TTG1* in *Arabidopsis*; however, we identified no mutations in the two *CqTTG1-like* genes in the *rebc* mutants. This suggests that *CqTTG1-like* genes cannot function as *REBC*, or that they are hypostatic genes of *REBC*. To determine the molecular mechanism underlying EBC formation, it is necessary to elucidate the relationship between *REBC* and *CqTTG1-like* genes, as well as their detailed functions. To achieve this goal, it is necessary to establish a new method for transforming quinoa, which is a challenge using currently available methods.

REBC is localized to nuclei and chloroplasts and is involved in the formation of both EBCs and chloroplasts, suggesting that REBC translocates to nuclei and chloroplasts after translation. The function of TTG1, a trichome WD40 protein, changes depending on the specific factors involved in its transcription[35,36]. Hence, we postulate that REBC localizes to the nucleus or chloroplasts depending on the proteins to which it is bound. We showed that REBC simultaneously regulates the formation of EBCs and chloroplasts, although the biological significance of this regulation is unknown. Furthermore, EBCs have also been implicated in the protection of leaves from UV-B radiation[19]. This particular environmental stress is greater at higher elevation (as in the Andes, the region from which quinoa originates), where light intensity is greater than in the lowlands. The ETR, an index of chloroplast function related to photosynthesis, was lower in *rebc* mutants than in the WT under strong light radiation. This suggests that REBC is involved in chloroplast formation, promoting more efficient photosynthesis under strong light. Thus, it is possible that quinoa evolved simultaneous regulation of EBC formation and chloroplast formation via REBC as the most efficient mechanism to promotes photosynthesis while still protecting plants under severe environmental stress. This notion is further supported by the observation that *REBC* expression is regulated by light. Alternatively, regarding the simultaneous formation of EBC and chloroplasts, secondary factors might affect the formation of EBCs because of impaired chloroplast formation. Future studies should seek to further characterize the roles of REBC in the formation of EBCs and chloroplasts.

Surprisingly, we observed that mutations in *REBC* alone caused the *rebc* phenotype, a rare occurrence in allotetraploid species

such as quinoa. *rebc* mutants appeared in the M$_3$ generation due to dysfunctions of other homologs, as reported in a previous study[22]. In hexaploid wheat, mutant strains produced by EMS mutagenesis also appear in the M$_3$ generation[37]. These findings suggest that despite the presence of multiple homologs in a polyploid plant, there are a certain number of gene families for which only one functional gene is present.

In this study, we identified the gene involved in EBC formation and revealed novel EBC functions. These findings provide insight into the roles of EBCs and the molecular mechanisms underlying their formation in *Chenopodium* spp., including quinoa. Furthermore, our observations will help elucidate the mechanism of stress tolerance in halophytes containing EBCs.

## Methods

**Plant materials, growth conditions, and mutagenesis**. Seeds of the CQ127 variety of quinoa were obtained from the USDA (Supplementary Table 7). Seeds of *Chenopodium* spp. were obtained from the USDA and the Institute of Plant Science and Resources at Okayama University (Supplementary Table 7). Seeds of the *Arabidopsis ttg1–21* (CS2105595) mutant on the Columbia (Col) background were obtained from the Arabidopsis Biological Resource Center. Quinoa seeds were sown in cell trays and grown at 23 °C under a neutral photoperiod (12-h light/12-h dark) in a phytotron. After 3 weeks, the seedlings were transplanted into standard potting mix (Ikubyou Baido, Takii, Kyoto, Japan) in 5 L plant pots and grown in a glasshouse. *Arabidopsis* plants were grown at 23 °C under long-day conditions (16-h light/8-h dark).

EMS mutagenesis was conducted by Inplanta Innovations (Yokohama, Japan); we obtained approximately 8000 mutagenized seeds (M$_1$ seeds) of the CQ127 variety. The mutagen-treated seeds were sown to produce M$_1$ progeny, which were propagated to obtain the M$_3$ generation. The M$_3$ progeny were then screened for mutants related to EBC formation.

**Microscopic analysis and EBC counts**. EBCs were examined using an Axiovert 200 optical microscope (Zeiss, Jena, Germany), and images were captured using the Axiovision 4.6 software (Zeiss). To enumerate EBCs, the abaxial sides of leaves from 2-week-old quinoa plants were photographed under a microscope. EBCs were counted using image data, and the number of EBCs per leaf was calculated.

Chlorophyll autofluorescence in EBCs was examined using an LSM 510 META confocal microscope (Zeiss). Images were captured and analyzed using LSM Image Examiner (Zeiss).

**Transmission electron microscopy**. Leaves were cut into sections (1.0–1.5 mm × 3 mm) using a razor blade immersed in 2.5% glutaraldehyde in 0.05 M cacodylate buffer (pH 7.2), and fixed for 18 h at 4 °C. The samples were then fixed with 1% OsO$_4$ in the same buffer for 12 h at 4 °C, rinsed in the same buffer for 10 min, dehydrated in ethanol, transferred to QY-1, and embedded in a Quetol 651 resin mixture (Nissin EM, Tokyo, Japan). Transverse sections (0.5-μm thickness) for light microscopy were obtained using an ultramicrotome (EM UC6; Leica, Vienna, Austria). Sections were dry-mounted on glass slides and stained with 1% (w/v) toluidine blue in 1% (w/v) sodium borate solution for 2–3 min at 90 °C. The sections were then examined by light microscopy, and areas of interest were selected for electron microscopy. Thin sections were cut using a diamond knife, collected on 200 × 75 mesh Formvar-coated grids, and stained with saturated uranyl acetate and lead citrate in 50% ethanol for 10 min. Specimens were viewed using an H-7650 transmission electron microscope (Hitachi, Tokyo, Japan).

**Osmium maceration and SEM**. Leaves were cut into sections (2 mm × 4 mm) using a razor blade and subjected to osmium maceration as reported in a previous study[38]. The specimens were initially fixed at 20 °C for 2 h in 1% osmium tetroxide

(OsO₄) solution buffered at pH 7.2 with 0.05 M cacodylate. After rinsing with buffer solution, the specimens were successively immersed in 5%, 30%, and 50% dimethyl sulfoxide (DMSO) solutions for 30 min per treatment. Specimens in 50% DMSO were frozen on an aluminum plate chilled with liquid nitrogen and split using a razor blade and hammer in a freeze-fracture apparatus (TF-1, EIKO Engineering, Tokyo, Japan). The split pieces were immediately placed in 50% DMSO solution at room temperature and thawed. The samples were then repeatedly rinsed in buffer (0.05 M cacodylate; pH 7.2) until the DMSO was completely removed, and then transferred to 0.1% OsO₄ buffered at pH 7.2 with 0.05 M cacodylate. The specimens were incubated at 20 °C for 9 days, and the OsO₄ solution was changed daily.

Following the 9-day incubation period, the specimens were again fixed in 1% OsO₄ for 1 h. The specimens were then rinsed with buffer solution and treated with 2% tannic acid for 1 h. The specimens were rinsed with distilled water, placed in 1% OsO₄ for 1 h to increase their electrical conductivity, and then dehydrated in a graded ethanol series. Following transfer to isoamyl acetate, the specimens were dried in a critical point dryer (HCP-1; Hitachi Koki, Tokyo, Japan) and coated with approximately 1 nm platinum in an ion coater (E-102, Hitachi). Metal-coated specimens were observed via field emission SEM (S-4700, Hitachi) at 25 kV.

**Abiotic stress treatments**. Four-week-old quinoa and *Arabidopsis* plants were subjected to wind stress treatment. A fan was used to generate a consistent wind current with a speed of 20–30 cm/s that struck the quinoa shoot apex from above throughout the treatment period of 4 weeks. After completing the stress treatment, the degree of damage to the shoot apex was evaluated and compared with that in untreated plants.

One-month-old quinoa plants were irradiated with fluorescent UV-B lamps (T-15M, Vilber Lourmat, France) with a cellulose acetate film to filter out UV-C. A distance of approximately 30 cm was maintained between the tops of the plants and the UV-B lamp. UV-B intensity was measured with a Solarmeter Model 6.2 UVB (Solar Light Company, Glenside, PA, USA). UV-B treatments were conducted at 1.5 W/m² for 4 h. Two weeks after treatment, the degree of damage to the shoot apex was evaluated and compared with that in untreated plants.

One-month-old quinoa plants were subjected to salt stress treatment by adding 2 L of 0.4 M NaCl solution on days 1 and 3, and 4 L of 0.6 M NaCl solution over 9 days (Fig. 4h). An equal amount of water was added to the control group (non-treated) on the same days as the salt treatments (Fig. 4h). Three weeks after salt treatment, the plants were photographed and the fresh weights of the shoots were measured. The Na⁺ contents of the tissue containing the leaf lamina were measured using a LAQUAtwin-Na-11 (Horiba, Kyoto, Japan). To investigate the accumulation of salt on EBCs in salt stress treatment, a paintbrush was used to remove EBCs from leaves under a microscope, and the Na⁺ contents of leaf lamina with or without EBCs in the same leaf were measured (Supplementary Fig. 2a). Accumulation of salt was expressed as relative Na⁺ content, with the value in a non-brushed leaf was defined as 100%. The fourth to sixth expanded leaves from the top of the plant subjected to the salt treatment were used for this experiment. Soil electrical conductivity was measured using a LAQUAtwin-B-771 (Horiba). The soil electrical conductivities of the salt-treated and untreated plots in this experiment were 48.20 and 0.82 dS/m, respectively.

One-month-old quinoa plants were subjected to heat treatment at 42 °C for 10 days. Two weeks after heat treatment, the plants were photographed, and the fresh weights of the shoots were measured.

Three-week-old plants were subjected to progressive drought treatment by withholding water for 6–12 days. Then, the plants were watered again and grown for a further 4 days before survival rates were determined. WT plants exposed to drought for more than 10 days were severely damaged. Therefore, we withheld water for 10 days for the evaluation of drought-stress tolerance in WT and mutant plants. Drought treatment was confirmed by the expression of drought stress–induced genes (*CqHSP20* and *CqNCED3*)[39]. Leaves of similar developmental stages taken before and after drought treatment were used to measure electrolyte leakage, as described in a previous report[40].

**Sample preparation for candidate gene identification**. To isolate the gene affected in the *rebc1* mutant, a heterozygous line of the *rebc1* mutant was grown to the four-leaf stage for phenotyping. Twenty-five plants of each of the WT and *rebc* phenotypes were pooled, and genomic DNA was extracted using the DNeasy Plant Mini Kit (Qiagen, Hilden, Germany). Pooled DNA from the WT and *rebc1* mutants was sequenced. Paired-end reads of 150 bp from each of the two pools were obtained using the Illumina HiSeq X System; sequencing was performed by Macrogen Japan (Kyoto, Japan). The reads used in this study are available from the DDBJ/EMBL/NCBI under the accession numbers DRX138189 (*rebc1* mutant pool) and DRX138190 (WT pool).

**In silico subtraction method**. Low-quality reads and adaptors in short reads from the *rebc1* pool were trimmed using Trimmomatic-0.32[41] with the following settings: SLIDINGWINDOW:10:20 LEADING:15 TRAILING:10 MINLEN:40. Reads from the WT pool were trimmed using Trimmomatic 0.32 with the following settings: HEADCROP:1 SLIDINGWINDOW:4:20 LEADING:20 TRAILING:20 MINLEN:40. Adaptor sequences were CACGACGCGTCTTCCGATCT and

ACCGCTCTTCCGATCTGTAA. WT-specific short reads were identified using customized Python scripts (https://github.com/Comai-Lab/kmer-extract-by-trigger-site)[26,27] with k = 37. On the basis of WT-specific short reads, we assembled scaffolds using Velvet 1.2.10[28]. A homology search of these scaffolds and predicted genes in the Quinoa Genome DataBase[20] was conducted using BLASTN[42], and scaffolds homologous to the predicted genes with a blast score > 200 were identified. Next, we removed scaffolds that were similar to transposable elements. Burrows–Wheeler Aligner (BWA) 0.7.12[43] alignments were conducted using the bwa aln command with the setting –n 1 and the bwa samse command with the settings. Nucleotide sites at which the WT carried heterozygous WT and mutant alleles and the mutant pool carried homozygous mutant alleles were detected using SAMtools tview 1.4.1[44].

**Identification of candidate mutations by the MutMap method**. To filter out low-quality short reads, we excluded reads for which >10% of the sequenced nucleotides had a Phred quality score <30. Candidate single-nucleotide polymorphisms (SNPs) were then analyzed using MutMap pipeline ver. 1.4.4 (http://genome-e.ibrc.or.jp/home/bioinformatics-team/mutmap)[24,25]. In the MutMap pipeline, alignment was conducted using BWA[43], and alignment files were converted to SAM/BAM files using SAMtools[44]. In this pipeline, short reads obtained from the WT pool were aligned to the mutant pool reference sequence, which was developed by replacing nucleotides of the public quinoa reference genome[20] with SNPs (SNP index > 0.9) detected by aligning the short reads obtained from the mutant pool. The SNP indexes at all SNP positions were then calculated for the WT pool.

**Molecular cloning**. Total RNA was extracted using the RNeasy Plant Mini Kit (Qiagen) and treated with RNase-free DNase I (Qiagen) to eliminate genomic DNA. First-strand cDNA was synthesized from 500 ng of total RNA using the Takara RNA PCR Kit (AMV) Ver. 3.0 (Takara Bio, Kusatsu, Japan) with oligo(dT) primers. Genomic DNA was extracted using the DNeasy Plant Mini Kit (Qiagen). We obtained the full-length open reading frame sequences of *REBC* (XM_021859495), *REBC1-like1* (XM_021883470), *REBC-like2* (XM_021859173), *CqTTG1-like1* (XM_021907203), and *CqTTG1-like2* (XM_021869994) from the NCBI gene database.

**Transient complementation of the *rebc* mutant**. To construct a plasmid for transient complementation experiments, the 2.5-kbp upstream region and the open reading frame region of the *REBC* gene were amplified by polymerase chain reaction (PCR). For binding of the amplified fragments, the primers used for amplification were designed to overlap by 20 bp with the fragments to be linked (Supplementary Table 8). The amplified PCR fragments were introduced into pCAMBIA1380 (CAMBIA, Canberra, Australia) by Gibson assembly (NEB, Ipswich, MA, USA). The resultant plasmid was introduced into *Rhizobium rhizogenes* (ATCC 15834) by electroporation[45] using Gene Pulser Xcell (Bio-Rad, Hercules, California, USA) to prepare a transformant. To prepare a control transformant, a plasmid expressing AcGFP1 was introduced into *R. rhizogenes*. Quinoa seeds were germinated on MS plates containing 4.3 g/L MS (Fujifilm, Tokyo, Japan), 1.5% sucrose, and 2.5 g/L Gelrite (Fujifilm) and kept at 23 °C under continuous light conditions for 2 weeks, the time required for *rebc* phenotype confirmation. Infection of quinoa with transformed *R. rhizogenes* was conducted as described previously[22].

**Generation of transgenic *Arabidopsis* plants**. PrimeSTAR GXL DNA polymerase (Takara Bio) and oligonucleotides containing a restriction enzyme cleavage site were used for PCR amplification (Supplementary Table 8). The amplified fragments were digested with the appropriate restriction enzymes and then introduced into the binary vector pCAMBIA1301MdNcoI[22]. An ABI PRISM 3100 genetic analyzer (Applied Biosystems, Foster City, CA, USA) was used to sequence the resultant plasmids. To generate transgenic *Arabidopsis* plants, binary plasmids were introduced into a *ttg1-21* mutant of *Arabidopsis* by *Agrobacterium tumefaciens*–mediated transformation using the floral dip method[46]. Transgenic plants were selected on a 0.8% (w/v) agar MS medium containing 50 μg/mL hygromycin. For each transgenic line, 10 or more independent plants were produced. Homozygous T₃ plants were used for further analysis.

**Reverse transcription (RT)-PCR analysis**. The High-Capacity cDNA RT Kit (ThermoFisher Scientific, Waltham, MA, USA) with random primers was used to synthesize first-strand cDNA from 500 ng of total RNA. RT-PCR was conducted using GeneAtlas 322 (Astec, Shime, Japan) with Taq DNA Polymerase (NEB). The procedure for amplification of the candidate transcripts consisted of initial denaturation at 94 °C for 2 min, followed by 35 cycles of 94 °C for 30 s, 55 °C for 30 s, and 72 °C for 1.5 min. *CqMON1* and *AtEF1α* were used as internal controls for relative gene expression in quinoa and *Arabidopsis*, respectively. Primer pairs are listed in Supplementary Table 8.

**Stereo-structural analysis**. Secondary structure prediction suggested that the REBC protein has a β-rich conformation (http://www.compbio.dundee.ac.uk/

jpred/). Because the amino acid sequence alignment initially identified no candidate structure for approximately 100 of the N-terminal residues of REBC, the three-dimensional structure was modeled for residues 102–448. F-box/WD-repeat protein 1A (PDB Code, 1p22) was used as the initial structure for the modeling. The three-dimensional structure of the REBC protein was predicted using Modeler ver. 9.16[47] and Phyre2[48] and displayed using MOLMOL[49].

**Immunoblot analysis**. To produce the REBC protein in *E. coli*, PrimeSTAR GXL DNA polymerase (Takara Bio) and oligonucleotides containing a restriction enzyme cleavage site were used to conduct PCR amplification (Supplementary Table 8). The amplified fragments were digested with the appropriate restriction enzymes, and then introduced into the expression vector pCold-TF DNA (Takara Bio). An ABI PRISM 3100 genetic analyzer (Applied Biosystems) was used to sequence the resultant plasmids. Trigger factor (TF)–REBC fusion protein and TF–rebc1 fusion protein were produced according to the manufacturer's instructions. Crude extract for immunoblot analysis was prepared using PBS (pH 7.4).

To detect REBC protein in quinoa, crude extract was prepared from quinoa leaves by grinding the leaves into a fine powder in liquid nitrogen, followed by sonication in extraction buffer [PBS with 0.1% Tween 20, 0.1% Triton X-100, and complete EDTA-free protease inhibitor cocktail (Roche, Basel, Switzerland)]. The resultant cell extracts were centrifuged at 20,000 × *g* for 10 min at 4 °C, and the collected supernatants were used for subsequent analysis.

An aliquot of crude extract (5 μL of *E. coli* extract and 10 μL of plant extract) was added to loading buffer (60 mM Tris-HCl, pH 6.8, 2% SDS, 5% 2-mercaptoethanol, and 5% glycerol). After boiling for 5 min, the proteins were analyzed by sodium dodecyl sulfate polyacrylamide gel electrophoresis (10% gel) and visualized using Coomassie Blue R250 (Bio-Rad). Separated proteins were transferred to an Immobilon-P transfer membrane (Merck Millipore, Burlington, MA, USA). The membrane was blocked by incubation in Tris-buffered saline containing 0.3% Tween 20 (TBS-T) and 1% (w/v) non-fat dry milk for 1 h, and then washed three times with TBS-T. The blocked membrane was then incubated for 1 h with anti-SNGDDDTEDTNSAG polyclonal rabbit IgG, (Scrum, Tokyo, Japan) (1:2000 dilution for *E. coli* extract and 1:500 dilution for plant extract) in blocking solution and washed. For visualization, horseradish peroxidase-conjugated anti-rabbit IgG (H+L chain) polyclonal antibody (MBL, Nagoya, Japan) was used at a dilution of 1: 5000. Chemi-Lumi One Ultra (Nacalai Tesque, Kyoto, Japan) was used for detection. Fluorescence images were obtained using a LAS3000 image analyzer (Fujifilm).

**Immunoelectron microscopy**. Cut leaves and shoot apices (1.0–1.5 mm × 2–3 mm) were fixed with 4% paraformaldehyde and 0.5% glutaraldehyde in 50 mM sodium cacodylate buffer (pH 7.2) for 1 h at 4 °C. The specimens were dehydrated in an ethanol series and embedded in LR white. Ultra-thin sections were cut with a diamond knife and mounted on formvar-coated nickel grids. Sections were incubated with anti-SNGDDDTEDTNSAG polyclonal rabbit IgG and then reacted with 15 nm gold-conjugated goat anti-rabbit IgG (BBI Solutions, Crumlin, UK). After immunolabeling, sections were stained with uranyl acetate and lead citrate. As a cytochemical control, specimens were incubated without primary antibody or with non-immune rabbit IgG. Samples were visualized by TEM (H-7650, Hitachi).

**Phylogenetic tree of deduced amino acid sequences**. The deduced amino acid sequences of REBC, REBC-like1, REBC-like2, CqTTG1-like1, and CqTTG1-like2 were aligned using the ClustalW algorithm[50] with TTG1 and REBC isolated from quinoa, beet (*Beta vulgaris*), spinach (*Spinacia oleracea*), amaranth (*Amaranthus hypochondriacus*), and *Arabidopsis*). Bootstrap values shown at nodes were obtained from 5000 trials. The phylogenetic tree was constructed using the neighbor-joining algorithm in the MEGA7 software[51].

**RNA-seq analysis**. RNA-seq analysis was conducted to determine comprehensive mRNA expression in the WT and *rebc* mutants. Pools were prepared from 10 individuals of the *rebc* phenotype, and the WT phenotype that were isolated from the *rebc1* and *rebc2* heterozygous lines. RNA was extracted from each pooled sample using the RNeasy Plant Mini Kit (Qiagen). For Illumina sequencing, 1 μg of RNA was used to prepare libraries according to the protocol for the NEBNext Poly (A) mRNA Magnetic Isolation Module (NEB). The libraries were subjected to 250 cycles of paired-end sequencing on MiSeq (Illumina, San Diego, California, USA). The sequence reads were filtered for quality in the FASTAQ format. The RNA-seq data from other groups (Experiment No. RX3124330–RX3124333) was obtained from the NCBI Sequence Read Archive (https://www.ncbi.nlm.nih.gov/sra). The reads were aligned to the Cq_PI614886_genome_V1_pseudomolecule reference sequence (http://www.cbrc.kaust.edu.sa/chenopodiumdb/) by HISAT2[52]. After genes were predicted by StringTie[53] using alignment data from HISAT2, the expression levels of each gene in each sample were compared using feature-Counts[54]. Genes with a total of ≤40 reads from four samples were discarded. Next, genes for which RNA expression was downregulated to less than half or upregulated more than twofold in both *rebc* mutants were selected (Supplementary Datas 1 and 2). Subcellular localization of the selected gene products was predicted using WolF PSORT (https://wolfpsort.hgc.jp/) (Supplementary Datas 1 and 2). Gene expression was compared with RNA-seq performed by other groups by

converting to TPM values[32]. Genes for which the TPM value was down- or upregulated by ≥ twofold in both *rebc* mutants were selected and added to the TPM value in EBCs (Supplementary Datas 3 and 4).

**Chloroplast analysis**. Chlorophyll from 3-week-old plants was extracted with 100% dimethylformamide, and the absorbances of supernatants were measured at 646.8 and 663.8 nm. Chlorophyll content was calculated as previously described[55].

Chlorophyll fluorescence parameters, i.e., the maximal quantum yield of fluorescence (Fv/Fm) and the light response curves of ETR, were measured with a JUNIOR-PAM fluorometer (Heinz-Walz Instruments, Effeltrich, Germany) and calculated as described previously[56]. For measurement of Fv/Fm, plants were dark-adapted for 30 min before measurements were made.

**Statistics and reproducibility**. Student's *t* test was computed to analyze the significant difference between two groups. Chi-squared test was used for segregation test. Asterisk indicated values differ significantly with the value at <0.05.

**Reporting summary**. Further information on research design is available in the Nature Research Reporting Summary linked to this article.

## Data availability

RNA-seq data is deposited in GenBank (Accession nos. DRX163715-163718, Supplementary Table 5). All other data generated or analyzed during this study are included in this published article and its Supplementary information files. Source data for figures is available in Supplementary Data 5.

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

## Acknowledgements
This work was supported by a co-operative research grant from Actree Co. The authors thank Akiko Mizuno, Hiroko Hayashi, Mami Awatani, and Fumino Shimada for their excellent technical assistance.

## Author contributions
M.M. and H.M. conceived this study. T.I., K.N., and M.M. designed the experiments and characterized the *rebc* mutant. H.K. performed the electron-microscopic analyses. H.T., Y.Y., N.M., and A.A. performed the next-generation sequencing analyses. S.O. performed the simulation analysis. T.I. and M.M. wrote the paper. All authors approved the final paper.

## Competing interests
The authors declare no competing interests.
