## [Peer Review File · Communications Biology]

Reviewers' comments:

Reviewer #1 (Remarks to the Author):

Quinoa, as a halophyte, is typically characterized by the epidermal bladder cells (EBCs) that form on the surface of leaves. It is well-known that EBCs have a capacity to accumulate high concentration of salt, which confers increased tolerance of plants to high salinity. Currently, the molecular mechanisms underlying the formation of EBCs in quinoa are still unknown. In this paper, using EMS-mediated mutagenesis of quinoa, the authors identified a gene, named REBC, that is required for the formation of EBCs. REBC encodes a WD40 protein, which is localized in both nucleus and chloroplast. Mutation of REBC gene also leads to a deficiency in the protection of shoot apex from abiotic stress. Although the authors discover a gene that is required for the formation of EBCs in quinoa, no experiments were performed to investigate the biological function of REBC in the regulation of the formation of EBCs. The writing needs to be improved.

Major points:

1. The mechanism underlying the role of REBC gene in the formation of EBCs needs to be investigated.
2. The pictures shown in Fig.1 are not novel. Similar data have already been presented in other papers.
3. Fig2 and Fig3, please also show the morphological phenotypes of *rebc2* mutant.
4. Fig3, the plants without water-mist or wind treatment should be included as a control. The authors should explain how water-mist and wind treatments damage shoot apex in *rebc* mutant. What is the association of this damage with reduction of EBCs?
5. Fig4 c-g, quantification of the number of EBCs?
6. Fig5, immunoblotting is suggested to examine the protein of REBC in the wild type and *rebc* mutants.
7. line 130, please explain how heterozygous *rebc* plants were generated?
8. line 144, what is the mean of "a partial gene".
9. What is the phenotype of *rebc* mutants under high salinity?
10. Whether the deficiency of chloroplast formation is associated with the reduction of EBCs in *rebc* mutants?

Reviewer #2 (Remarks to the Author):

The paper by Imamura et al. ask questions about the gene(s) involved in the formation of epidermal bladder cell in quinoa.

Bladder cells represent modified trichomes. TTG1 a WD40 protein has been shown in Arabidopsis to govern trichome formation. However, the key genes involved in EBC formation have not yet been isolated yet.

To determine the genetic basis of their formation, the authors performed a mutant screen and obtain two mutants with reduced EBCs (*rebc*) and abnormal chloroplasts. The mutation of REBC encodes a WD40 protein, the authors localized in the nucleus and chloroplasts. From finding normal environmental conditions already damaging mutant plants the authors conclude that EBCs play a role in tolerating nature.

Given that quinoa is seen as health food and increase in production and stress tolerance represents a current breeding aim, new insights bladder development is a topic the general reader of CB wishes find.

Comments and questions

Introduction:

Statement 1: Trichomes can protect plants from both biotic and abiotic stresses¹, stomata undergo water transpiration and gas exchange, and root hairs aid moisture and nutrients

absorption by increasing the root surface area.

Comment: The reader is confused! Why mentioning the role of stomata out of any trichome context? And later on, stomata are not mentioned anymore. Thus, the stomata part of the sentence should be removed.

Results:

Statement 2: Because EBCs are densely packed around and over the shoot apex (Fig. 2 and Supplementary Fig. 1), we hypothesized that they play a role in protecting the apex from abiotic stress. To test this hypothesis, we sprayed water mist over the shoot apices ..

And: Other stressors, including wind ..

Comment 1: reader has not heard about water mist and wind stress. Water mist and wind is what plants face in nature. The authors need to elaborate on this issue.

Comment 2: The paper Kiani-Pouya et al. (cited 12) documents that bladderless quinoa grows not different from WT under control conditions, but suffers from soil salinity. Thus, mutant testing salt, drought stress, and UV light is a must, because EBCs were associated with tolerance against these stressors and this is the reader gets from the authors introduction.

Statement 3: REBC mRNA was expressed in the leaves, hypocotyls, and roots of quinoa seedlings

..

Comment: authors should mention whether or not there mutants show root hair phenotype?

Statement 4: Immunoelectrolysis was performed using an antibody against the REBC C-terminal tail to determine the subcellular localization of the REBC protein (Fig. 5a and Supplementary Fig. 5). The REBC protein was localized to the nuclei and chloroplasts in the leaves and to the nuclei in the shoot apices (Fig. 5c-l).

Comment 1: together with this statement the reader (in the main text) would like to learn about the specificity of the antibody in order to judge localization in nucleus and chloroplasts.

Comment 2: from what the reader understood a chloroplast phenotype was shown for one allele only. It has to be shown for the other allele too, to exclude the possibility that the phenotype results mutation outside the rebc gene.

Comment 3: The reader is not told whether leaf cell and bladder chloroplast share the same abnormalities. And how these abnormalities feedback on chloroplast function such gross photosynthesis and UV susceptibility of PS electron transport.

Statement 5: RNA sequencing (RNA-Seq) analysis was performed to assess the gene expression profile of the rebc mutants (Supplementary Table 2). We identified genes that were upregulated at least 2-fold or downregulated to less than a 50% in the rebc mutants compared with the WT.

Comment: Papers Zou et al. and Bohm et al. cited identified genes differentially expressed in bladders. Thus, the reader would like to know how much an overlap with the rebc mutant profile exists.

Discussion:

Statement 6: Trichomes are similarly densely packed around the shoot apices of in Arabidopsis (Supplementary Fig. 9), suggesting that they may also provide protection to the shoot apex in that species.

Comment: In this context the reader needs to learn how Arabidopsis WT and trichome mutants respond to water mist and wind.

Statement 7: In addition to the protective effects of EBCs observed here, these cells have also been implicated in the protection of young quinoa leaves from UV-B radiation³⁵. This particular environmental stress in the Andes, the region from which quinoa originates,

Comment: Given the reader again is told that bladder protect from UV stress, it is a must to test for the UV mutant phenotype.

Statement 8: This study identifies novel EBC functions and the gene involved in their formation. ..., this information may be applicable to the development of methods promoting crop growth in harsh environments.

Comment: latter statement about EBC function is not correct given that before the authors mention correctly "... we have shown that REBC simultaneously regulates the formation of EBCs and chloroplasts, although the biological significance of this regulation is unknown."

Minor comments:

- Abb 2d: blurry und low resolution

- Why was the screen done in the M3 and not in the M2 generation?
- Abb 6K too small letter
- -Abb 6 too blurry
- Line 130 heterologous or do they mean heterozygous?

Reviewer #3 (Remarks to the Author):

In this work, Imamura and co-authors employed ethylmethanesulfonate mutagenesis approach to understand mechanisms of epidermal bald cell (EBCs) formation in quinoa. The authors report that mutation of REBC gene which encodes a WD40 protein localizes in the nucleus and chloroplasts plays a critical role in EBC formations. Contrary to expectations, the reported REBC protein differed from TTG1, which is a WD40 protein involved in trichome formation, suggesting that these two pathways are largely independent. They also show a protective role of EBC against the apex damage by mechanical stimulation and mist, in phenotyping experiments. The reported results are highly novel and of a significant interest to the journal readership.

Being rather positive about the overall approach and the novelty of reported data, I have found the phenotyping results being at par with reported molecular data. The choice of abiotic stresses (water mist and wind) is very strange if not odds. Of course, any structure that provide a mechanical protection will prevent stress-induced damage to young tissues from these factors. This does not explain, however, neither the appearance of EBC on more advanced leaves, neither the reasons of why plants invest into development of EBC rather than trichomes (with will come with much lower carbon cost to plants). Thus, to convince the reader about the important role of EBC in preventing the damage of root apex by abiotic stresses, the authors should test the range of more "traditional" stresses such as drought, UV or extreme temperatures. Also, the reported evidence should include not only photographs and the number of damaged plants but also some quantitative physiological data. In this context, and given the fact that the reported genes are localized in chloroplasts, I am very surprised of why some very basic characteristics such as chlorophyll content of Fv/Fm chlorophyll fluorescence ration are not reported here? This is odds...

The paper will also benefit for the more in-depth interpretation on the possible difference in EBC and trichome formation and, specifically, evolutionary and adaptive significance of such differentiation.

Responses to the comments of Reviewer 1

Major points:

1. The mechanism underlying the role of REBC gene in the formation of EBCs needs to be investigated.

Although the investigation of the mechanism of *REBC* in the formation of EBCs would be useful and of great interest, we agree with the editor's opinion that such an undertaking is beyond the scope of the current study, and thus, did not conduct further functional analysis on the *REBC* gene for this publication. Regarding future pursuit of this work, however, we have compared the RNA-seq data of this study with the RNA-seq for EBC tissues reported by other groups, and we listed gene candidates involved in EBC formation (Supplementary Tables 7 and 8).

2. The pictures shown in Fig. 1 are not novel. Similar data have already been presented in other papers.

As the reviewer points out, most of the images in Fig. 1 are not new. However, in this paper, we have retained Fig. 1 for readers who may not know EBCs, thinking that the existence of Fig. 1 would enhance their understanding.

3. Fig2 and Fig3, please also show the morphological phenotypes of *rebc2* mutant.

In accordance with this suggestion of the reviewer #1, we have added a picture of the *rebc2* mutant at the point indicated (Fig. 2, Fig. 3, Fig. 4, and Supplementary Fig. 2).

4. Fig3, the plants without water-mist or wind treatment should be included as a control. The authors should explain how water-mist and wind treatments damage shoot apex in *rebc* mutant. What is the association of this damage with reduction of EBCs?

Per this suggestion of reviewer #1, we have added control (non-treated) to all abiotic stress experiments (Fig. 3, Fig. 4, and Supplementary Fig. 2).

We have added the following text explaining how wind stress and water-mist stress damage the shoot apex (lines 120–128).

“In outdoor cultivation environments, wind, rain, and dew are environmental stresses that intermittently affect plants as they grow. These stresses can directly affect and damage the shoot apex from the outside. For quinoa growing in harsh environments, protecting the shoot apex, which is more vulnerable to environmental stress than other

tissues due to the presence of many immature cells, against these stresses is particularly important. Because EBCs are densely packed around and over the shoot apex (Fig. 2d, g, and Supplementary Fig. 1), we hypothesized that EBCs play a role in protecting the apex from these stresses. Under this hypothesis, *rebc* mutants with reduced EBCs are expected to have reduced resistance to these stresses.”

5. Fig4 c-g, quantification of the Su of EBCs?

Following the suggestion of reviewer #1, we counted EBCs and listed the number of EBCs per leaf in Supplementary table 1. We have also added relevant sentences to the corresponding sections (lines 113–114, 189, and 278–279).

6. Fig5, immunoblotting is suggested to examine the protein of REBC in the wild type and *rebc* mutants.

Per this suggestion of reviewer #1, we performed western blotting on wild-type and *rebc1* mutant (Supplementary Fig. 6b). We have added the following sentence to the corresponding sections in Methods:

“Crude extract was prepared using phosphate-buffered saline (PBS, pH 7.4) for immunoblot analysis.

To detect REBC protein in quinoa, crude extract was prepared from quinoa leaves by grinding them to a fine powder in liquid nitrogen, followed by sonication in extraction buffer [PBS with 0.1% Tween 20, 0.1% Triton X-100, and complete EDTA-free protease inhibitor cocktail (Roche)]. The resultant cell extracts were centrifuged at 20,000× g for 10 min at 4°C, and the collected supernatants were used for subsequent analysis.” (lines 588–594).

7. line 130, please explain how heterozygous *rebc* plants were generated?

Thank you for pointing this out. The heterozygous *rebc* plants show the parent (M2) of the *rebc* mutants. We have revised the text “heterologous *rebc* plants” as “the parent line (M2) of *rebc* mutants” (lines 155–156).

8. line 144, what is the mean of “a partial gene”.

Thank you for pointing this out. “A partial gene” has been revised to “gene”. The corresponding sentence was revised to “The reference genome registered in the quinoa genome database has low accuracy because it is a draft genome. Therefore, when a BLAST search was performed on Cqu_c00398.1_g001.1 in the NCBI database, in which a highly accurate quinoa genome was registered, only one gene (XM_021859495) was found.” (lines 168–172).

9. What is the phenotype of *rebc* mutants under high salinity?

Following this suggestion of reviewer #1, WT and *rebc* mutants were cultured under high salinity conditions and their growth was evaluated. The growth of the *rebc* mutant was more inhibited than that of the WT under high salinity conditions (Fig. 4). We have added sentences explaining this to the corresponding sections in the Results (lines 141–147), Methods (lines 465–472), and Figure Legends (lines 849–856)

10. Whether the deficiency of chloroplast formation is associated with the reduction of EBCs in *rebc* mutants?

This study suggests that REBC protein is localized in chloroplasts and nuclei, suggesting that EBC formation and chloroplast formation may function independently. However, this possibility that the reviewers pointed out remains. Further studies will elucidate the mechanism underlying EBC formation. A corresponding sentence was added: “Additionally, regarding the simultaneous formation of EBC and chloroplasts, another possibility is that secondary factors owing to impaired chloroplast formation may affect the formation of EBCs. Further analysis of the formation of EBCs and chloroplasts by REBC proteins is needed.” (lines 363–366).

The writing needs to be improved.

We have revised the English using the English language review service of Enago (www.enago.jp).

Responses to the comments of Reviewer 2

Comment on Statement 1: The reader is confused! Why mentioning the role of stomata out of any trichome context? And later on, stomata are not mentioned anymore. Thus, the stomata part of the sentence should be removed.

Thank you for pointing this out. Following this suggestion of reviewer #2, we have removed the description of stomata from the manuscript.

Comment 1 on Statement 2: reader has not heard about water mist and wind stress. Water mist and wind is what plants face in nature. The authors need to elaborate on this issue..

Following this suggestion of the reviewer #2, we have added explanations for wind stress and water mist stress (lines 120–129).

“In outdoor cultivation environments, wind, rain, and dew are environmental stresses that intermittently affect plants as they grow. These stresses can directly affect and damage the shoot apex from the outside. For quinoa growing in harsh environments, protecting the shoot apex, which is more vulnerable to environmental stress than other tissues due to the presence of many immature cells, against these stresses is particularly important. Because EBCs are densely packed around and over the shoot apex (Fig. 2d, g, and Supplementary Fig. 1), we hypothesized that EBCs play a role in protecting the apex from these stresses. Under this hypothesis, *rebc* mutants with reduced EBCs are expected to have reduced resistance to these stresses. To test this hypothesis, we sprayed water mist that reproduces rain and dew over the shoot apices of WT and *rebc* mutant plants.”

Comment 2 on Statement 2: The paper Kiani-Pouya et al. (cited 12) documents that bladderless quinoa grows not different from WT under control conditions, but suffers from soil salinity. Thus, mutant testing salt, drought stress, and UV light is a must, because EBCs were associated with tolerance against these stressors and this is the reader gets from the authors introduction.

To address this suggestion of reviewer #2, we performed salt, drought, and UV stress experiments (Fig. 3, Fig. 4, and Supplementary Fig. 2). We have added text to the corresponding sections in the Results (lines 137–151), Methods (lines 457–472 and lines 478–485), and Figure Legends (lines 844–856)

Comment on Statement 3: authors should mention whether or not there mutants show root hair phenotype?

Following this suggestion of reviewer #2, we observed the root hairs of WT and mutants (Supplementary Fig. 5b). The following sentence has been added: “Although *REBC* expression was confirmed in the roots, no significant difference was observed in roots and root hair traits between WT and *rebc* mutants (Supplementary Fig. 5b).” (lines 199–201)

Comment 1 on Statement 4: together with this statement the reader (in the main text) would like to learn about the specificity of the antibody in order to judge localization in nucleus and chloroplasts.

Following this suggestion of reviewer #2, we performed western blotting for wild-type and *rebc1* mutants (Supplementary Fig. 6b). The corresponding sentence has been added: “To determine the subcellular localization of the REBC protein, we developed an antibody against the REBC C-terminal tail and confirmed specificity for REBC protein in *E.coli* and plant cells” (lines 220–222), and Methods (lines 588–595 and 604).

Comment 2 on Statement 4: from what the reader understood a chloroplast phenotype was shown for one allele only. It has to be shown for the other allele too, to exclude the possibility that the phenotype results mutation outside the *rebc* gene.

Per this suggestion of reviewer #2, we have added photographs of chloroplasts in *rebc2* mutant (Fig. 8 and Supplementary Fig. 9).

Comment 3 on Statement 4 (first sentence): The reader is not told whether leaf cell and bladder chloroplast share the same abnormalities.

Following this suggestion of reviewer #2, we observed chloroplasts in EBC for WT and *rebc* mutants by transmission electron microscopy (TEM). EBC chloroplasts present in WT could be observed; however, regarding the EBC observation of the *rebc* mutants, compared with the WT, it was very difficult to observe the chloroplasts from the EBC that were barely present. We could only obtain TEM image data of EBC chloroplasts in *rebc2* mutant (Supplementary Fig. 10d, e). Therefore, as an additional experiment, we observed chlorophyll autofluorescence to evaluate the chloroplast traits

in EBC. As a result, it was revealed that the *rebc* mutants also caused abnormalities in the chloroplasts of EBC (Fig. 8k-r).

We have added text to the corresponding sections in the Results (lines 276–289), Discussion (line 294), Methods (lines 407–408), and Figure Legends (lines 909–914)

Comment 3 on Statement 4 (second sentence): And hoe these abnormalities feedback on chloroplast function such gross photosynthesis and UV susceptibility of PS electron transport.

Following this suggestion of reviewer #2, we measured Fv/Fm and electron transport rate (ETR) to study chloroplast abnormalities in *rebc* mutants. We observed no difference in Fv/Fm between WT and *rebc* mutants (Fig. 8c). For ETR measurement, it was clarified that the ETR of both *rebc* mutants was lower than that of WT under high-light radiation, whereas no difference was observed between WT and *rebc* mutants under weak-light radiation (Fig. 8d). This measurement confirmed that the *rebc* mutant had abnormal PSII electron transfer under high-light conditions. We have added sentences to the corresponding sections in the Results (lines 265–272), Discussion (lines 356–360), Methods (lines 658–662), and Figure Legends (lines 900–907)

Comment on Statement 5: Papers Zou et al. and Bohm et al. cited identified genes differentially expressed in bladders. Thus, the reader would like to know how much an overlap with the *rebc* mutant profile exists.

Following the suggestion of reviewer #2, we obtained public data and performed a comparative analysis. First, a group was selected in which the expression in *rebc* mutants was reduced by more than twice or less than 1/2 in the TPM value compared with the wild type. Next, the TPM value was calculated from the RNA-seq data of EBCs from the public data and added to the list of genes whose expression fluctuated in *rebc* mutants (Supplementary Tables 7 and 8). We added text to the corresponding sections in the Results (lines 250–260), Discussion (lines 333–334) and Methods (lines 637–639 and lines 647–651).

Comment on Statement 6: In this context the reader needs to learn how Arabidopsis WT and trichome mutants respond to water mist and wind.

In compliance with the suggestion of reviewer #2, we performed abiotic stress treatments (mist and water treatment) on wild-type and *rebc1* mutants in *Arabidopsis* (Supplementary Fig. 11).

Based on the results of the stress experiment, we revised the text as follows: “Although not as dense as quinoa, trichomes similarly existed around the shoot apices in *Arabidopsis* (Supplementary Fig. 11). As a result of the same stress experiments as performed in quinoa, no difference was observed between the WT and the *ttg1* mutant in *Arabidopsis*, suggesting that trichomes do not play a role in protecting shoot apices from environmental stress (Supplementary Fig. 11). Thus, shoot apex protection from environmental stress may be one of the unique functions of EBCs.” (lines 314–319).

Comment on Statement 7: Given the reader again is told that bladder protect from UV stress, it is s must to test for the UV mutant phenotype.

Wind and water mist stress were performed on *Arabidopsis* WT and *ttg1* mutants, as in quinoa. Damage was not found in *ttg1* as in WT. (Supplementary Fig. 11). Based on this result, we concluded that the protection of shoot apices by EBCs is a unique function not found in trichomes. Therefore, analysis using the *Arabidopsis* UV mutant was not performed.

Comment on Statement 8: latter statement about EBC function is not correct given that before the authors mention correctly “.., we have shown that REBC simultaneously regulates the formation of EBCs and chloroplasts, although the biological significance of this regulation is unknown.”

As per these comments of the reviewer, we have deleted the corresponding sentence.

Minor comments:

- **Abb 2d: blurry und low resolution**

To comply with this, photographs have been replaced with high-quality images.

- **Why was the screen done in the M3 and not in the M2 generation?**

Because no mutant for EBCs was obtained in the M2 generation, screening was performed in the M3 generation.

- **Abb 6K too small letter**

To facilitate readability in compliance with this comment, we have increased the font of the phylogenetic tree.

- **Abb 6 too blurry**

Photographs have been replaced. Fig. 7 as well as have modified Supplementary Fig. 8 along with this modification.

- **Line 130 heterologous or do they mean heterozygous?**

We have revised the text “heterologous *rebc* plants” to “the parent line (M2) of *rebc* mutants.” (lines 155–156).

Responses to the comments of Reviewer 3

The choice of abiotic stresses (water mist and wind) is very strange if not odds.

We added the following text explaining why wind stress and water-mist stress damage the shoot apex (lines 120–126).

“In outdoor cultivation environments, wind, rain, and dew are environmental stresses that intermittently affect plants as they grow. These stresses can directly affect and damage the shoot apex from the outside. For quinoa growing in harsh environments, protecting the shoot apex against these stresses is particularly important. Because EBCs are densely packed around and over the shoot apex (Fig. 2d, g, and Supplementary Fig. 1), we hypothesized that EBCs play a role in protecting the apex from these stresses. Based on this hypothesis, *rebc* mutants with reduced EBCs are expected to have reduced resistance to these stresses.”

Of course, any structure that provide a mechanical protection will prevent stress-induced damage to young tissues form these factors. This does not explain, however, neither the appearance of EBC on more advanced leaves, neither the reasons of why plants invest into development of EBC rather than trichomes (with will come with much lower carbon cost to plants).

To comply with the suggestion of reviewer #3, we have added the following comment (lines 319–324):

“In quinoa, it is better to cover the shoot apex completely with EBCs than to cover it with a trichome that creates a gap. Ensuring that the shoot apex is protected even at the expense of extra energy for EBC formation is important for quinoa growing in harsh environments.”

Thus, to convince the reader about the important role of EBC in preventing the damage of root apex by abiotic stresses, the authors should test the range of more “traditional” stresses such as drought, UV or extreme temperatures. Also, the reported evidence should include not only photographs and the number of damaged plants but also some quantitative physiological data.

Following this suggestion of reviewer #3, we performed UV-B, high salinity, drought, and high-temperature stress experiments (Fig. 3, Fig. 4, and Supplementary Fig. 2). We also analyzed some quantitative physiological data and added it to the corresponding sections. We have added corresponding text to the Results (lines 137–151), Methods (lines 457–485), and Figure Legends (lines 844–856).

In this context, and given the fact that the reported genes are localized in chloroplasts, I am very surprised of why some very basic characteristics such as chlorophyll content of Fv/Fm chlorophyll fluorescence ration are not reported here? This is odds....

Per this suggestion of reviewer #3, we measured Fv/Fm and chlorophyll contents to investigate chloroplasts in WT and *rebc* mutants (Fig. 8b, c). We have added corresponding text to the Results (lines 265–269), Discussion (lines 356–360), Methods (lines 653–662), and Figure Legends (lines 900–907).

Reviewers' comments:

Reviewer #1 (Remarks to the Author):

Here are some further comments:

1. Line 119-128, no evidence shows that the EBCs around the shoot apex is evolved to protect the damage from environmental stresses, and the results presented in this study could not support the conclusion that the enhanced damage of shoot apices in the rebc mutant is associated with the less number of EBCs.
2. Have authors detected the expression of REBC in the shoot apices?
3. line 209, "No mutations were observed in the regions homologous to the rebc mutants" is not clear.
4. line 210, it is unclear how authors concluded that "Therefore, we predict that the primary function of REBC protein is EBC formation" based on the analysis of the paralogs of REBC.
5. Line 259, some DEGs are expressed in the EBCs does not mean that these genes are involved in EBCs formation, and we don't know how many genes are specifically expressed in EBCs?
6. The language needs to be further improved.

Reviewer #2 (Remarks to the Author):

The results obtained from the additional experiments requested by the referees clearly improved the paper.

From what I can see most of the referees' questions have been answered properly. However, the authors may wish to answer the 5 open questions listed below.

Statement 1: Recently, some candidate genes related to EBC formation in *M. crystallinum* have been selected^{12, 13}. Because the key genes involved in EBC formation have not yet been isolated, the molecular mechanism underlying the formation of these important cells remains unknown.
Comment: After mentioning *M. crystallinum* key genes involved in EBC formation in the introduction the reader would like to see them discussed in relation the DEGs identified in the quinoa rebc MT.

Statement 2: In outdoor cultivation environments, wind, rain, and dew are environmental stresses that intermittently affect plants as they grow.

Comment: The reader does not agree dew being classified as environmental stress. All in vitro cultured plant material is facing high humidity and dew without stress phenotypes reported in publication. How does quinoa WT and rebc MT perform in invitro culture?

Statement 3: Furthermore, WT and rebc mutants were evaluated under conditions of drought stress (withholding water for five days) .. . No difference was found between the WT and rebc mutants under the treatment conditions of this study for either drought stress ..

Comment: Quinoa is known for its water stress tolerance. How the authors know that withholding water for five days quinoa experienced drought stress? Under drought stress ABA is produced and ABA genes get expressed. The authors' statement about drought stress is only justified if they had tested for well-known stress markers.

Statement 4A: .. transient complementation experiments .. Rhizobium rhizogenes in quinoa rebc

mutants. Infection of mutant plants with a line carrying a plasmid expressing WT REBC resulted in the formation of a few EBCs on leaves ..

Statement 4B: REBC cannot complement the *ttg1* phenotype. In addition, *Arabidopsis* does not contain an ortholog of REBC. These findings suggest that EBC formation has a different molecular mechanism from trichomes.

Comment: Given that REBC expression did rescue the *At GLABRA* MT indicates that *rebc* is not inducing trichome formation in non-quinoa plants.

How does transient *rebc* overexpression affect bladder density in quinoa WT?

Statement 5: RNA-seq has been reported for expression in quinoa EBCs9, 10. Using the published data, we .. . In this group, some genes showed particularly high expression in EBCs, such as MLP-like protein (Supplementary Table 7). In addition, 57 out of 92 genes whose .. . Thus, some of the genes obtained by this analysis may be involved in EBC formation.

Comment: The lines of evidence in this paragraph the reader cannot follow. Please explain the reader, how MLP-like protein and 57 out of 92 genes should contribute to bladder formation.

Reviewer #3 (Remarks to the Author):

The authors did a good job revising their work and addressing all critical comments made by reviewers. The paper has been greatly improved and reads well now. As a result, I have only one (relative minor) request to make. The authors have conducted some additional experiments treating plants with NaCl. The reported data is shows that plants lacking EBC show more salt sensitive phenotype. This is consistent with the very recent paper in the filed reporting a positive correlation between EBC volume and salinity stress tolerance in quinoa, based on screening of over 100 quinoa accessions (Kiani-Pouya et al 2019 *Env Exp Bot*). However, the reported Na content data (Fig 4t) is confusing and shows no difference between WT and mutants. I believe this abnormality may be explained by the fact that the authors have analysed Na content of INTACT leaves, without separating leaf lamina from EBC. This is wrong, as it does not allow to differentiate between two fractions. Such experiments should be repeated, and Na content should be compared between leaf lamina only (e.g. after the brushing off EBC). In this case, I would predict a significantly lower amounts of Na accumulated in the leaf mesophyll of plant bearing EBC (as compared with mutants).

A very minor comment: In 468-469. Please report Na content in the soil as EC of the saturated paste extract. The currently used unit (Na in dry soil) is meaningless.

Responses to the comments of Reviewer 1

1. Line 119-128, no evidence shows that the EBCs around the shoot apex is evolved to protect the damage from environmental stresses, and the results presented in this study could not support the conclusion that the enhanced damage of shoot apices in the *rebc* mutant is associated with the less number of EBCs.

According to the reviewer's comments, we have revised these sentences (lines 118–120, 136–137). Moreover, we have deleted the description of EBCs protection form the manuscript.

2. Have authors detected the expression of REBC in the shoot apices?

Thank you for your suggestion. We have investigated the *REBC* expression in shoot apices and young ear using RT-PCR. These results were included as Supplementary Fig. 9. We have also revised the corresponding sentence in the manuscript (line 208).

3. line 209, “No mutations were observed in the regions homologous to the *rebc* mutants” is not clear.

Thank you for pointing this out. We have deleted this sentence.

4. line 210, it is unclear how authors concluded that “Therefore, we predict that the primary function of REBC protein is EBC formation” based on the analysis of the paralogs of REBC.

Thank you for highlighting this point. We have deleted these sentences.

5. Line 259, some DEGs are expressed in the EBCs does not mean that these genes are involved in EBCs formation, and we don't know how many genes are specifically expressed in EBCs?

As pointed out by the reviewer, most of the genes expressed in EBCs are not involved in EBC formation. In this study, we obtain a quinoa ortholog (jasmonate-induced

protein family) that is suggested to be involved in the EBC formation of *M. crystallinum*, as observed when the genes of the EBC-less mutant were compared with those of the *rebc* mutants. Based on this result, it is considered that the jasmonate-induced protein family gene selected in quinoa may be involved in quinoa EBC formation. We have added relevant information in the corresponding sections of the revised manuscript (lines 265–271).

6. The language needs to be further improved.

Our manuscript has been edited for English language and grammar by Enago (<https://www.enago.jp/>).

Responses to the comments of Reviewer 2

Comment Statement 1: After mentioning *M. crystallinum* key genes involved in EBC formation in the introduction the reader would like to see them discussed in relation the DEGs identified in the quinoa *rebc* MT.

According to the reviewer's comments, we have compared the genes whose expression was altered in the *rebc* mutants of quinoa and in the EBC-less mutants of *M. crystallinum*. We have added following sentences in the result section: "The genes with altered expression in *rebc* mutants were compared with those that showed altered expression in a mutant form of *M. crystallinum* that lacked EBCs^{13, 14}. A gene belonging to the jasmonate-induced protein family was decreased in both *M. crystallinum* (WM28; NCBI Acc. No. KT366265) and quinoa (Phytozome Acc. No. AUR62022156) (Supplementary Table 7). It has been reported that the heterologous expression of WM28 in *Arabidopsis* increases the number of trichomes¹³. Therefore, the quinoa jasmonate-induced protein gene might be involved in EBC formation in quinoa." (lines 265–271).

Comment Statement 2: The reader does not agree dew being classified as environmental stress. All in vitro cultured plant material is facing high humidity

and dew without stress phenotypes reported in publication. How does quinoa WT and rebc MT perform in in vitro culture?

Following this suggestion, we have examined the *in vitro* water-mist treatment and included the relevant information in the revised manuscript (lines 120–124, 126–133). We have also included details in the Methods sentence (lines 463–467).

Comment Statement 3: Quinoa is known for its water stress tolerance. How do the authors know that withholding water for five days quinoa experienced drought stress? Under drought stress ABA is produced and ABA genes get expressed. The authors' statement about drought stress is only justified if they had tested for well-known stress markers.

Following this suggestion, we have conducted a drought stress test again using the expression of drought-stress-response gene in quinoa (*CqHSP20*, *CqNCED3*) reported by Morales et al (2017). In the 6-week-old plant used in the previous experiment, expression of the marker gene was observed in the untreated plant. Therefore, 3-week-old plants without expression of the marker gene in untreated plants were used for the drought stress test. These results were replaced with the previous results in Supplementary Fig 5. We have added relevant sentences to the corresponding text in the Results (lines 156, 159), Methods (lines 502–509), and References (lines 787–789) sections.

Comment Statement 4AB: Given that REBC expression did not rescue the At GLABRA MT indicates that rebc is not inducing trichome formation in non-quinoa plants. How does transient rebc overexpression affect bladder density in quinoa WT?

Thank you for your suggestion, we examined the transient *REBC* overexpression in quinoa WT and measured the density of EBCs per leaf. In the transient overexpression of *REBC* in quinoa WT, the density of EBCs did not increase (Supplementary Fig 8). This result indicates that overexpression of *REBC* in WT

may not affect the density of EBCs. We have also added relevant sentences to the corresponding text in the revised manuscript (lines 201–204).

Comment on Statement 5: The lines of evidence in this paragraph the reader cannot follow. Please explain the reader, how MLP-like protein and 57 out of 92 genes should contribute to bladder formation.

Thank you for pointing this out. We have deleted following sentence “ In this group, some genes showed particularly high expression in EBCs, such as MLP-like protein (Supplementary Table 7)”. We have added the results of comparison between the EBC-related genes in *M. crystallinum* (lines 265–271).

Responses to the comments of Reviewer 3

The authors have conducted some additional experiments treating plants with NaCl. The reported data is shows that plants lacking EBC show more salt sensitive phenotype. This is consistent with the very recent paper in the filed reporting a positive correlation between EBC volume and salinity stress tolerance in quinoa, based on screening of over 100 quinoa accessions (Kiani-Pouya et al 2019 Env Exp Bot).

Thank you for the information, we have added the recent findings (Kiani-Pouya et al 2019) to the introduction (lines 67–69) and reference (lines 714–716) sections.

However, the reported Na content data (Fig 4t) is confusing and shows no difference between WT and mutants. I believe this abnormality may be explained by the fact that the authors have analysed Na content of INTACT leaves, without separating leaf lamina from EBC. This is wrong, as it does not allow to differentiate between two fractions. Such experiments should be repeated, and Na content should be compared between leaf lamina only (e.g. after the brushing off EBC). In this case, I would predict a significantly lower amounts of Na

accumulated in the leaf mesophyll of plant bearing EBC (as compared with mutants).

According to the reviewer's instructions, we measured Na^+ concentration using the tissue from the leaf lamina instead of the shoot apex (Fig 4t, u). Accordingly, no difference in Na^+ concentration was observed between the WT and *rebc* mutants. Furthermore, Na^+ concentration was measured in leaves with and without EBCs in the salt-treated WT (Supplementary Fig 3). As a result, the salt concentration of the leaves brushed with EBCs was lower than that of untreated leaves (Supplementary Fig 3).

Based on the obtained results, the sentences "To investigate the accumulation on EBCs of salt in salt-treated quinoa, we compared the Na^+ content of leaves with/without EBCs. In the leaves from which EBCs were removed, the Na^+ content was reduced by approximately 10% compare with leaves with EBCs (Supplementary Fig. 3), suggesting that salt was accumulated in EBCs under this salt stress treatment. These results suggested that Na^+ accumulation in a cell of *rebc* mutants is higher than that in WT plants." were added to the Result section (lines 149–154). In addition, we have added a sentence to the text in the Methods (lines 488–493) and Legends (lines 880) sections.

Please report Na content in the soil as EC of the saturated paste extract. The currently used unit (Na in dry soil) is meaningless.

Accordingly, we have measured the soil electrical conductivity and changed from Na^+ concentration to soil electrical conductivity (lines 494–496).

Reviewers' comments:

Reviewer #1 (Remarks to the Author):

The authors have addressed all my concerns.

Reviewer #2 (Remarks to the Author):

A bladderless mutant was identified and associated with a defect in REBC gene.

Given that the bladder phenotype is associated with a chloroplast defect, one would have expected that another not identified mutation is involved.

The evidence presented for a role of REBC in abiotic stress protection in general and salt stress in particular is insufficient.

Reviewer #3 (Remarks to the Author):

I believe that the authors did a good job addressing the issues raised by all reviewers, and most my concerns have been dealt with. As such, I have only two additional comments to make:

1. I still feel very uncomfortable with the "mist stress". Such stress hardly exists in nature, and its impact is very difficult to quantify without the actual measurements of the partial oxygen pressure in the appropriate tissue. As this was not done, I suggest to simply omit the relevant data from the MS. The paper is interesting enough even without it.

2. The paper will benefit from another round of editing by a native English speaker. Some constructions could be improved for clarity. For example, ln 45 says "rebc mutants sustained damage to their shoot apices under abiotic stress...". Plants cannot "sustain damage"; the better term would be "displayed damage" or "had their apex damaged". Also, the word "stress" should be in plural. Such things are minor but important for the journal of this standing.

Responses to the comments of Reviewer #1

The authors have addressed all my concerns.

Thank you very much for your help in improving the manuscript.

Responses to the comments of Reviewer #2

A bladderless mutant was identified and associated with a defect in REBC gene.

Given that the bladder phenotype is associated with a chloroplast defect, one would have expected that another not identified mutation is involved.

rebc mutants were homozygous with a recessive phenotype caused by a single-gene mutation. In the *rebc* mutants, the traits of abnormal chloroplasts and reduced EBCs were not segregated.

We obtained two independent *rebc* mutants. The *REBC* gene, which is the causative gene associated with the *rebc* mutant phenotype, was identified using NGS. We have demonstrated that the *REBC* gene is the causative gene for the *rebc* mutant by general breeding techniques. In addition, immunoelectron microscopy revealed that REBC was localized in the nucleus and chloroplasts, suggesting that it functions in both EBCs and chloroplasts. These results indicate that the *rebc* phenotype is derived from mutations in the *REBC* gene.

Following the instructions of the editor and reviewers, we also have revised the corresponding sentences (lines 191, 255–257).

The evidence presented for a role of REBC in abiotic stress protection in general and salt stress in particular is insufficient.

The abiotic-stress treatments were performed using two independent *rebc* mutants and the wild type. In addition, regarding salt treatment, the results were the same as those of other groups. Therefore, we do not consider these findings to be experimentally insufficient.

Following the instructions of the editor and reviewers, we removed the water-mist treatment.

Responses to the comments of Reviewer #3

1. I still feel very uncomfortable with the “mist stress”. Such stress hardly exists in nature, and its impact is very difficult to quantify without the actual measurements of the partial oxygen pressure in the appropriate tissue. As this was not done, I suggest to simply omit the relevant data from the MS. The paper is interesting enough even without it.

Following the suggestion of the reviewer, we removed the water-mist treatment data and corresponding sentences.

2. The paper will benefit from another round of editing by a native English speaker. Some constructions could be improved for clarity. For example, ln 45 says “rebc mutants sustained damage to their shoot apices under abiotic stress...”. Plants cannot “sustain damage”; the better term would be “displayed damage” or “had their apex damaged”. Also, the word “stress” should be in plural. Such things are minor but important for the journal of this standing.

Our manuscript has been edited for English language and grammar by ZENIS
(<https://www.zenis.co.jp/eng/index.html>).